

# Improving the representation of high-latitude vegetation in
# Dynamic Global Vegetation Models
Peter Horvath[1, 4], Hui Tang[1,2, 4], Rune Halvorsen[1], Frode Stordal[2, 4], Lena Merete Tallaksen[4, 5],
Terje Koren Berntsen[2, 4], Anders Bryn[1, 3, 4]
[1] Geo-Ecology Research Group, Natural History Museum, University of Oslo, P.O. Box 1172, Blindern NO-0318
Oslo, Norway
[2] Section of Meteorology and Oceanography, Department of Geosciences, University of Oslo, Norway
[3] Division of Survey and Statistics, Norwegian Institute of Bioeconomy Research, P.O. Box 115, NO-1431 Ås,
Norway
[4] LATICE Research Group, Department of Geosciences, University of Oslo, Norway
[5] Section of Physical geography and Hydrology, Department of Geosciences, University of Oslo, Norway
Correspondence to: Horvath, P. (peter.horvath@nhm.uio.no)
**Keywords**: Area frame survey, Community Land Model, CLM4.5BGCDV, Distribution model, Earth System
Model, Plant functional types, Remote sensing, Vegetation types,





**Abstract**. Vegetation is an important component in global ecosystems, affecting the physical, hydrological and
biogeochemical properties of the land surface. Accordingly, the way vegetation is parameterised strongly
influences predictions of future climate by Earth system models. To capture future spatial and temporal changes
in vegetation cover and its feedbacks to the climate system, dynamic global vegetation models (DGVM) are
included as important components of land surface models. Variation in the predicted vegetation cover from
DGVMs therefore has large impacts on modelled radiative and non-radiative properties, especially over high-
latitude regions. DGVMs are mostly evaluated by remotely sensed products, but rarely by other vegetation
products or by in-situ field observations. In this study, we evaluate the performance of three methods for spatial
representation of vegetation cover with respect to prediction of plant functional type (PFT) profiles – one based
upon distribution models (DM), one that uses a remote sensing (RS) dataset and a DGVM (CLM4.5BGCDV).
PFT profiles obtained from an independently collected vegetation data set from Norway were used for the
evaluation. We found that RS-based PFT profiles matched the reference dataset best, closely followed by DM,
whereas predictions from DGVM often deviated strongly from the reference. DGVM predictions overestimated
the area covered by boreal needleleaf evergreen trees and bare ground at the expense of boreal broadleaf deciduous
trees and shrubs. Based on environmental predictors identified by DM as important, we suggest implementation
of three novel PFT-specific thresholds for establishment in the DGVM. We performed a series of sensitivity
experiments to demonstrate that these thresholds improve the performance of the DGVM. The results highlight
the potential of using PFT-specific thresholds obtained by DM in development and benchmarking of DGVMs for
broader regions. Also, we emphasize the potential of establishing DM as a reliable method for providing PFT
distributions for evaluation of DGVMs alongside RS.





## 1 Introduction

Vegetation plays an important role in the climate system, as changes in the vegetation cover alter the biogeophysical and biogeochemical properties of the land surface (Davin and de Noblet-Ducoudré, 2010; Duveiller et al., 2018). Therefore an accurate descriptions of the vegetation distribution hold a key role in Earth system models (ESM) (Bonan, 2016; Poulter et al., 2015). Historical and present vegetation distributions are implemented in ESMs by means of datasets prepared from satellite observations (Lawrence and Chase, 2007; Li et al., 2018; Lawrence et al., 2011). However, in order to predict the future temporal and spatial changes in natural vegetation cover and subsequently the processes, dynamics and feedbacks to the climate system, dynamic global vegetation models (DGVMs) are needed.

DGVMs have been implemented as components of ESMs (Bonan et al., 2003) to represent long-term vegetation changes by a set of parameterizations describing general physiological principles, including ecological disturbances, successions (Seo and Kim, 2019) and species interactions (Scheiter et al., 2013). DGVMs represent the heterogeneity of land surface processes and interactions with other components of the Earth system by characterising land areas by their composition of type units defined by plant functional types (PFTs) (Bonan et al., 2003; Oleson et al., 2013). PFTs are groupings of plant species with similar eco-physiological properties – which express differences in growth form (woody vs herbaceous), leaf longevity (deciduous vs evergreen) and photosynthetic pathway (C3 and C4) (Wullschleger et al., 2014). Even though the DGVMs are being constantly developed and improved to incorporate more complex plant processes (Fisher et al., 2010), there are still fundamental challenges for DGVMs to correctly simulate the extents of the PFTs that characterise boreal and Arctic ecoregions (Gotangco Castillo et al., 2012). For instance, the thematic resolution of high-latitude PFTs is still limited (Wullschleger et al., 2014), important interactions between vegetation and fire in high latitudes are still missing (Seo and Kim, 2019), and forest carbon storage in the high latitude is still underestimated by most DGVMs (Song et al., 2013). The large uncertainties in simulating high-latitude PFT distributions may also lead to discrepancies between modelled and observed energy fluxes and hydrology (Hartley et al., 2017) or carbon cycles (Sitch et al., 2008). Accordingly, systematic evaluation of PFT distributions modelled by DGVMs is required to improve the DGVMs and, subsequently, to reduce uncertainties in estimates of climate sensitivity and in predictions by ESMs.

Remote sensing (RS) is often used for evaluation, benchmarking and improvement of parameters in of DGVMs (Zhu et al., 2018). RS products are commonly used to describe vegetation cover using vegetation classes derived from multispectral images based on vegetation indices such as the normalized difference vegetation index (NDVI) (Xie et al., 2008; Franklin and Wulder, 2002). For evaluation, RS products are translated into distributions of the PFT classes used in the DGVMs (Lawrence and Chase, 2007; Poulter et al., 2011). However, inconsistencies between various available RS-based land cover or vegetation products have been reported (Myers-Smith et al., 2011) and benchmarking DGVMs only to these RS-based products may therefore lead to different conclusions in ESMs (Poulter et al., 2015).

Among the less explored methods to generate wall-to-wall vegetation cover predictions is distribution modelling. Distribution models (DMs) are most often used to predict the distribution of a target, by establishment of statistical relationship between the target (response) and the environment (predictors) (e.g. Halvorsen, 2012). The most common use of DM in ecology is for prediction of species distributions (Henderson et al., 2014), but DM methods have proved valuable also for prediction of targets at higher levels of bio-, geo- or eco-diversity (i.e. vegetation



types and land-cover types) (Ullerud et al., 2016; Horvath et al., 2019; Simensen et al., accepted). DM methods
are inherently static, in contrast to the dynamic DGVMs (Snell et al., 2014). Nevertheless, they may be a useful
corrective to DGVMs by providing insights into important environmental factors driving the distribution of
individual targets, which may, in turn, improve PFT parameter settings in DGVMs.
Comparative studies that evaluate the present-day PFT distributions of DGVMs in a systematic manner, with
reference to a field-based evaluation dataset, are so far lacking. In this study, we evaluate representations of
vegetation, translated to PFT profiles, obtained by the three different methods (DGVM, RS, DM). We use an
independently collected field-based dataset (AR; the Norwegian National map series for Area Resources) for the
evaluation. Furthermore, we explore if environmental correlates of vegetation-type distributions identified by DM
can be used to improve DGVMs by adjusting parameter settings for high-latitude PFTs.
To approach these aims, we constructed a conversion scheme to harmonize the classification schemes of RS, DM
and AR into the PFTs used by the DGVM. We represent the vegetation coverage by using plant functional type
profiles (PFT profiles), vectors of relative abundances of PFTs within an area, e.g. given study plot, summing to
1. We then compare the PFT profiles obtained by DGVM, RS and DM with the AR reference on 20 selected study
plots across the Norwegian mainland. Finally, we conduct a series of sensitivity experiments to explore if DGVM
performance can be improved by adjusting DGVM parameters for selected environmental drivers.

## 2  Methods

### 2.1  Study area – Norway

The study area covers mainland Norway, spanning latitudes from 57°57'N to 71°11'N and longitudes from 4°29'E
to 31°10'E. Norway is characterized by a gradient from a rugged terrain with deep valleys and fjords in the western,
oceanic parts to gently undulating hills and shallow valleys in the central and eastern, more continental parts.
Temperature and precipitation show considerable variation with latitude, distance from the coast and altitude
(Førland, 1979). While the mean annual precipitation ranges from 278 mm in the central inland of S Norway to
more than 5000 mm in mid-fjord regions along the western coast, the yearly mean temperature ranges from 7°C
in the southwestern lowlands to –4°C in the high mountains (Hanssen-Bauer et al., 2017).
The vegetation of Norway is structured along two main climatic gradients; related to temperature/growing-season
length and humidity/oceanity (Bakkestuen et al., 2008). Broadleaf deciduous forests, regularly found in the
southern and southwestern parts (the boreonemoral bioclimatic zone), are further west and north (in the southern
boreal zone) restricted to locally warm sites (Moen, 1999). With declining temperatures northwards and towards
higher altitudes, i.e. in the southern and middle boreal zones, evergreen coniferous boreal forests dominate. In the
northern boreal zone they pass gradually into subalpine birch forests which form the tree line in Norway. A total
of about 38% of mainland Norway is covered by forests, and about 37% of the land is situated above the forest
line (of which two thirds is covered by alpine mountain heaths). Wetlands cover approximately 9% and broadleaf
deciduous forests about 0.4% of the land area (Bryn et al., 2018).

### 2.2  The AR reference dataset

Data obtained by in-situ field mapping, which is considered among the most reliable sources of land-cover
information (Alexander and Millington, 2000), is practically and economically impossible to obtain for large land





areas such as countries (Ullerud et al., 2020). Area-frame surveys based upon stratified statistical sampling may,
however, provide accurate, area-representative, homogeneous and unbiased land-cover and land-use data for large
areas. To evaluate the three methods for representing vegetation addressed in this study, we used the 'Norwegian
land cover and land resource survey of the outfields' (*Arealregnskap for utmark*) dataset (Strand, 2013), a
Norwegian implementation of the mapping program LUCAS (Eurostat, 2003). Data were collected in the period
between 2004–2014 in a systematic 18×18 km grid of 1081 rectangular plots (each 0.6×1.5 km, i.e. 0.9 km$^2$) (Bryn
et al., 2018; Strand, 2013). In each plot, expert field surveyors performed land-cover mapping by use of a system
with 57 land-cover and vegetation-type classes (Bryn et al., 2018), mapped at a scale of 1:25 000. The data were
provided in vector format with vegetation-type attributes assigned to each mapped polygon.
**2.3    Study plots**
Twenty out of the 1081 rectangular AR plots were selected to make up our reference dataset, AR (Fig. 1; center
coordinates in Table S1). The AR plots spanned elevations from 88 to 1670 m a.s.l., with mean annual temperatures
between -4.0°C and 7.1°C and mean annual precipitation between 466 and 2661 mm (Table S1). A test showed
that the selection of plots were acceptable representative for bioclimatic variation in Norway (see Fig. S3 and Fig.
S4). The test was performed using gridded temperature and precipitation data from seNorge2 (Lussana et al.,
2018a; Lussana et al., 2018b), interpolated for each plot by kriging in accordance with Horvath et al. (2019).

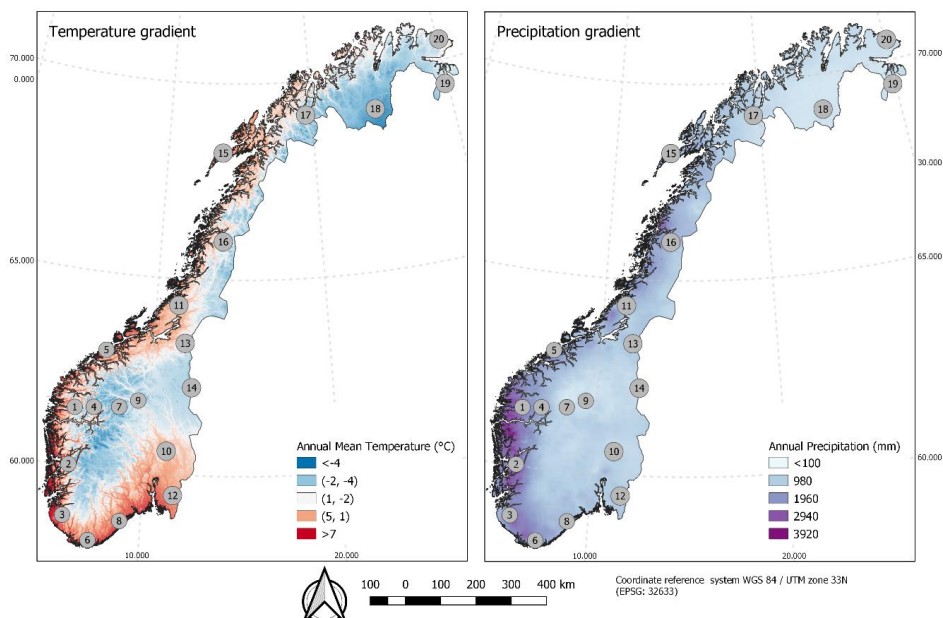

**Figure 1 - Locations of the 20 plots across the two main bioclimatic gradients in the study area: temperature (left) and**
**precipitation (right). The plots are numbered by longitude from west to east. Exact values of temperature, precipitation**
**and altitude for each plot are given in Table S1.**





**2.4    Methods for representing vegetation**
In this study, we use 'plot' as a collective term for two partly overlapping spatial units: (i) the 0.9-km² rectangles
of the AR of the reference dataset; and (ii) the 1-km² quadrats with the same centerpoint as, and edges parallel to
those of, the AR rectangles. The latter were used for the three methods of DGVM, RS and DM (Fig. S2).
Representations of the vegetation of each of these 20 plots were obtained by three different methods: (i) as the
result of single-cell DGVM simulations for each plot; (ii) inferred from an RS vegetation map of the study area;
and (iii) from vegetation-type DM models (Table 1). In order to make the three methods comparable, vegetation
was represented by plant functional type profiles (PFT profiles), obtained by a conversion scheme (Table 2 and
Sect. 2.5). We define a PFT profile as a thematic representation of the land surface in a given plot or a group of
plots, described as a vector of relative PFT abundances, i.e. values that sum up to 1.
**Table 1 – Details of each of the presented methods for representing vegetation. DGVM – dynamic global vegetation**
**model, RS – remote sensing, DM – distribution model. PFT – plant functional type, VT – vegetation type.**

|  | DGVM | RS | DM |
|---|---|---|---|
| Model type | Process-based mechanistic model | Supervised and unsupervised classification | Statistical model |
| Software / model name and version | Community Land Model 4.5 – CLM4.5-BGCDV | ENVI (image analysis) and ArcGIS (classification) | R version 3.6.2, generalized linear model |
| Reference | Oleson et al., 2013 | Johansen, 2009 | Horvath et al., 2019 |
| Thematic resolution | 14 PFTs | 25 VTs | 31 VTs |
| Spatial resolution (grid cell) | 1 km | 30 m | 100 m |


**2.4.1    The DGVM method**
The DGVM employed in this study was the CLM4.5BGCDV (further referred to as DGVM) embedded in NCAR's
Community Land Model version 4.5 (CLM4.5) (Oleson et al., 2013). In DGVM, plant photosynthesis, stomatal
conductance, carbon/nitrogen allocation, plant phenology and multi-layer soil biogeochemistry are described in
accordance with default CLM4.5, while vegetation dynamics (establishment, survival, mortality and light
competition) are handled separately based upon relatively simple assumptions (Oleson et al., 2013). We used
DGVM in the form of single-cell simulations for the 20 plots with grid-cell size set to 1×1 km (Table 1) to simulate
the fractional cover of each PFT. All models were run with default CLM4.5 values for surface parameters (e.g.
soil texture and depth), with prescribed atmospheric forcing derived from the 3-hourly hindcast of the regional
model (SMHI-RCA4) for the European Domain of the Coordinated Downscaling Experiment – CORDEX for
1980–2010 (Dyrrdal et al., 2018). The CORDEX model simulation was used because it has a higher spatial
resolution than the default atmospheric forcing used in CLM4.5 (0.11°×0.11° and 0.5°×0.5°, respectively). An
inspection of the choice of atmospheric forcing, by which the CORDEX data were compared with the SeNorge
data used for DM, showed minimal differences (Fig. S5). Only results obtained using CORDEX data are therefore
shown in this paper.
The model was run with default PFT parameters (Table S6). Among the 15 PFTs used in CLM4.5 to represent
vegetated surfaces globally(Lawrence and Chase, 2007), only six (plus bare ground) were relevant for our study
area (Table 2). Bare ground was predicted to occur where plant productivity was below a threshold value
(Dallmeyer et al., 2019). The DGVM simulates the vegetated landunit only (non-grey boxes in Fig. S7) while other
landunits within the 20 plots, including glaciers, wetlands, lakes, cultivated land and urban areas, make up the
"EXCL" PFT category (Table 2). We obtained PFT profiles for each plot by excluding the EXCL category and
recalculating fractions of the vegetated land unit covered by each PFT. Each model simulation was spun-up for





400 years to establish a vegetation in equilibrium with the current climate after initialization from bare ground. A
20-year average at the end of the spin-up was used as input for calculation of PFT profiles.

### 2.4.2  The RS method

As RS product we used SatVeg (Johansen, 2009), a vegetation map for Norway with 25 land-cover classes and a
spatial resolution (pixel size) of 30 m (Table 1). SatVeg is obtained by a combination of unsupervised and
supervised classification methods, applied to Landsat 5/TM and Landsat 7/ETM+ images within the near-infrared
and mid-infrared spectrum. Only pixels that were within each 1-km$^2$ plot with majority of their area were taken
into consideration for further calculations.

### 2.4.3  The DM method

The distribution models (DMs) for 31 vegetation types (VT) obtained by Horvath et al. (2019) using generalized
linear models (GLMs, with logit link and binomial errors, i.e. logistic regression), were used for this study. The
DMs were obtained by using wall-to-wall data for 116 environmental variables, gridded to a spatial resolution of
100×100 m (Table 1) as predictors. All DMs were evaluated by use of an independently collected data set (see
Horvath et al., 2019 for details). A seamless vegetation map (i.e. with one predicted VT for each pixel with no
overlap and no gaps) was obtained from the stack of 31 probability surfaces by assigning to each grid cell the VT
with the highest predicted probability of occurrence within that cell (Ferrier et al., 2002). Pixels that were within
each 1-km$^2$ plot with majority of their area were used for further calculations (Fig. S2).

### 2.5  Conversion to PFT profiles

Harmonisation of the various vegetation classification systems was accomplished by a conversion scheme that
represented each grid cell (RS and DM) or polygon (AR) in each of the 20 plots with one out of the six PFTs
recognised by DGVM (Table 2 and Fig. S2). The scheme was obtained by expert judgements and solicited by a
consensus process which involved ecologists participating in the AR18x18 survey as well as scientists working
with RS and DGVMs.
We used the conversion scheme of Table 2 to generate wall-to-wall PFT maps from the original RS, DM and AR
datasets (Table 1) by assigning one PFT to each 30×30 m grid cell, 100×100 m grid cell or VT polygon,
respectively. PFT profiles for each plot at the same thematic resolution as for DGVM were obtained as the vector
with fractions of grid cells or polygons assigned to each of the six PFTs. 'EXCL' classes not represented in DGVM
(cf. Table 2) were left out in order to minimise effects of land use, which could otherwise have brought about
differences in PFT profiles among the compared methods. PFT profiles were obtained for each combination of
method and plot. Aggregated PFT profiles were obtained by averaging the 20 PFT profiles obtained for each
method.

**Table 2– Conversion scheme for harmonizing vegetation and land cover types across methods (RS, DM and AR) into**
**plant functional types (PFTs). DGVM – dynamic global vegetation model, RS – remote sensing, DM – distribution**
**model. PFT – plant functional type, VT – vegetation type.**

| DGVM | | RS | DM | AR |
|---|---|---|---|---|
| PFT code | plant functional type | vegetation / land cover type – remote sensing | vegetation type – distribution model | vegetation type – area frame survey |





| | | | | |
|---|---|---|---|---|
| BG | Bare ground | Exposed alpine ridges, scree and rock complex | Frozen ground, leeward | Frozen ground, leeward |
| | | | Frozen ground, ridge | Frozen ground, ridge |
| | | | | Sand dunes and gravel beaches |
| | | | Boulder field | Pioneer alluvial vegetation |
| | | | Exposed bedrock | Barren land |
| | | | | Boulder field |
| | | | | Exposed bedrock |
| Boreal NET | Boreal needleleaf evergreen tree | Coniferous forest – dense canopy layer | Lichen and heather pine forest | Lichen and heather pine forest |
| | | Coniferous forest and mixed forest - open canopy | Bilberry pine forest | Bilberry pine forest |
| | | Lichen rich pine forest | Lichen & heather spruce forest | Meadow pine forest |
| | | | Bilberry spruce forest | Pine forest on lime soils |
| | | | Meadow spruce forest | Lichen & heather spruce forest |
| | | | Damp forest | Bilberry spruce forest |
| | | | Bog forest | Meadow spruce forest |
| | | | | Damp forest |
| | | | | Bog forest |
| Temperate BDT | Temperate broadleaf deciduous tree | Low herb forest and broadleaved deciduous forest | Poor / Rich broadleaf deciduous forest | Poor broadleaf deciduous forest |
| | | | | Rich broadleaf deciduous forest |
| Boreal BDT | Boreal broadleaf deciduous tree | Tall herb - tall fern deciduous forest | Lichen and heather birch forest | Lichen and heather birch forest |
| | | Bilberry- low fern birch forest | Bilberry birch forest | Bilberry birch forest |
| | | Crowberry birch forest | Meadow birch forest | Meadow birch forest |
| | | Lichen-rich birch forest | Alder forest | Birch forest on lime soils |
| | | | Pasture land forest | Alder forest |
| | | | Poor / rich swamp forest | Pasture land forest |
| | | | | Poor swamp forest |
| | | | | Rich swamp forest |
| Boreal BDS | Boreal broadleaf deciduous shrub | Heather-rich alpine ridge vegetation | Lichen heath | Lichen heath |
| | | Lichen-rich heathland | Mountain avens heath | Mountain avens heath |
| | | Heather- and grass-rich early snow patch communities | Dwarf shrub / Alpine calluna heath | Dwarf shrub heath |
| | | Fresh heather and dwarf-shrub communities (u/l) | Alpine damp heath | Alpine calluna heath |
| | | | Coastal heath / Coastal calluna heath | Alpine damp heath |
| | | | Damp heath | Flood-plain shrubs |
| | | | | Coastal heath |
| | | | | Coastal calluna heath |
| | | | | Damp heath |
| | | | | Crags and thicket |
| C3 | C3 grass | Graminoid alpine ridge vegetation | Moss snowbed / Sedge and grass snowbed | Moss snowbed |
| | | Herb-rich meadows (up-/lowland) | Dry grass heath | Sedge and grass snowbed |
| | | Grass and dwarf willow snow-patch vegetation | Low herb / forb meadow | Dry grass heath |
| | | | | Low herb meadow |
| | | | | Low forb meadow |
| | | | | Moist and shore meadows |
| EXCL | Excluded | Ombrotrophic bog and low-grown swamp vegetation | Bog / Mud-bottom fen and bog | Bog |
| | | Tall-grown swamp vegetation | Deer-grass fen / fen | Deer-grass fen |





| | | Wet mires, sedge swamps and reed beds | Sedge marsh | Fen |
|---|---|---|---|---|
| | | Glacier, snow and wet snow-patch vegetation | Pastures | Mud-bottom fen and bog |
| | | Water | | Sedge marsh |
| | | Agricultural areas | | Cultivated land |
| | | Cities and built-up areas | | Pastures |
| | | Unclassified and shadow affected areas, | | Built-up areas |
| | | | | Scattered housing |
| | | | | Artificial impediment |
| | | | | Glaciers and perpetual snow |
| | | | | Sea and ocean |
| | | | | Water bodies (fresh) |


### 2.6    Comparison of PFT profiles

Aggregated PFT profiles obtained by each of the DGVM, RS and DM methods were compared with the aggregated
PFT profile of the AR reference dataset by a chi-square test (Zuur et al., 2007).
For each plot, the dissimilarity between PFTs profiles obtained by each of the DGVM, RS and DM methods and
the PFT profile of the AR dataset was calculated by using proportional dissimilarity (Czekanowski, 1909):
$d_{hj}=\sum|y_{hji}-y_{0ji}|/\sum(y_{hji}+y_{0ji})=1-2\sum min(y_{hji},y_{0ji})/\sum(y_{hji}+y_{0ji})$
where $y_{hji}$ refers to the specific element in a PFT profile vector (the fraction occupied by the PFT in question) given
by method $h$ (DGVM, RS or DM; $h = 1, ..., 3$; the value $h = 0$ refers to the AR reference dataset), $j$ refers to
sampling unit ($j = 1, ..., 20$) and $i$ refers to PFT ($i = 1, ..., 6$). Proportional dissimilarity is the Manhattan measure
standardized by division by the sum of the pairwise sums of variable values (here PFTs). Since the values of each
PFT profile sums to one, the index reduces to
$d_{hj}=1-\sum min(y_{hji},y_{0ji})$
The proportional dissimilarity index is appropriate for incidence data like PFT abundances, i.e. variables that take
zero or positive values. The index reaches a maximum value of 1 when two objects have no common presences
(here, PFTs present in both compared objects) and ignore joint absences (zeros). We compared pairwise
differences between the proportional dissimilarity values among methods, using a Wilcoxon-Mann-Whitney
paired samples test.
All raster and vector operations related to DM, RS and AR were carried out in R (version 3.4.3) (R Core Team,
2019) using packages "rgdal" (Rowlingson, 2019), "raster" (Hijmans, 2019) and "sp" (Pebesma and Bivand,
2005), while graphics are produced using the "ggplot2" package (Wickham, 2016). Statistical analyses were
carried out in R (version 3.4.3), using the "vegan" package (Oksanen et al., 2019). All maps were produced in
QGIS (QGIS Development Team, 2019).

### 3    Results

The aggregated PFT profiles for the RS and DM datasets did not differ significantly from those of the reference
AR dataset according to the chi-square test, while a significant difference was found for the DGVM profiles (Table
3). While the proportion of pixels attributed to the PFT 'boreal NET' by the RS and DM methods underestimated
AR values by 3.0 and 2.8 percentage points, respectively, DGVM overestimated the proportion of boreal NET by





20.4 percentage points compared to the AR reference. Also, unproductive areas (BG) were overrepresented by
DGVM (by 16.6 percentage points), less so by RS (4.0 percentage points), while this PFT was slightly
underrepresented by DM (by 5.0 percentage points). Discrepancies were also observed for the cover of the C3
PFT, which was overestimated by RS and DM (by 7.2 and 2.9 percentage points, respectively) and underestimated
by 3.0 percentage points by DGVM. Furthermore, DGVM overestimated BG and temperate BDT cover on the
expense of boreal BDT and boreal BDS.
**Table 3 - PFT profiles (columns) aggregated across all 20 plots for the three methods compared in this study and the**
**AR reference dataset. Results of comparisons of aggregated PFT profiles for each of the three methods with the**
**reference are also given. DGVM – dynamic global vegetation model, RS – remote sensing, DM – distribution model, AR**
**– reference dataset. BG – bare ground, boreal NET – boreal needleleaf evergreen trees, temperate BDT – temperate**
**broadleaf deciduous trees, boreal BDT – boreal broadleaf deciduous trees; boreal BDS - boreal broadleaf deciduous**
**shrub, C3 – C3 grasses.**

| PFT | Compared methods | | | Reference |
|---|---|---|---|---|
| | DGVM (%) | RS (%) | DM (%) | AR (%) |
| BG | 29.5 | 17.0 | 7.9 | 12.9 |
| Boreal NET | 57.2 | 34.0 | 33.8 | 36.8 |
| Temperate BDT | 5.6 | 2.0 | 0.2 | 0.5 |
| Boreal BDT | 3.1 | 12.5 | 17.2 | 15.5 |
| Boreal BDS | 4.1 | 23.8 | 34.5 | 30.8 |
| C3 | 0.5 | 10.7 | 6.4 | 3.5 |
| Chi-square test | $\chi^2$= 45.98, df = 5, p < 0.05 | $\chi^2$= 6.36, df = 5, p = 0.27 | $\chi^2$= 2.61, df = 5, p = 0.75 | |


In accordance with results from comparisons between aggregated PFT profiles obtained by the three methods and
those obtained for the reference dataset, DGVM profiles for individual plots were significantly more dissimilar to
the AR reference than were RS and DM profiles (Fig. 2). While RS had the lowest median proportional
dissimilarity with the AR reference (0.19, compared to 0.26 for DM and 0.41 for DGVM), DM had the lowest
spread of dissimilarity values, measured as interquartile difference (0.12, compared to 0.19 for RS and 0.72 for
DGVM), among the three methods (Fig. 3). While no dissimilarity value for RS was above 0.50, two sampling
units (4, 19) acted as strong outliers in the distribution of DM values (cf. Fig. 2 and Fig. 3). A comparison of
proportional dissimilarity between pairs of methods revealed significant differences between DGVM profiles and
those obtained by RS and DM (Wilcoxon rank-sum tests: W = 111, p = 0.0167; and W = 88, p = 0.0026,
respectively), while RS and DM profiles were not significantly different from each other (Wilcoxon rank-sum test:
W = 161, p = 0.3013).


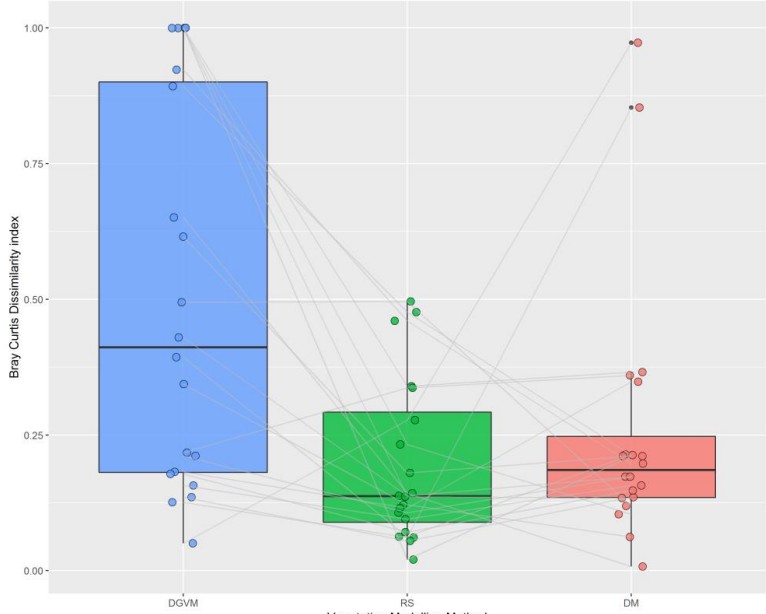


**Figure 2 - Proportional dissimilarity values between PFT profiles for each combination of 20 plots and one of the three methods compared in this study, and the corresponding plot in the AR reference dataset. The thick horizontal line, the box and the whiskers represent the median, the interquartile difference and the range of values for each method.**

Visual inspection of spatial patterns of PFT profile characteristics across the 20 plots suggests that the best agreement among the methods was obtained for the southeastern part of the study area, dominated by the boreal NET (Fig. 3). Compared to the AR reference dataset, PFT profiles obtained by DGVM were strongly biased: in the north (plots 17 and 18) towards boreal NET on the cost of boreal BDT, near the west coast (1, 2, 5 and 15) towards boreal NET on the cost of boreal BDS, and in southern coastal areas (3, 6 and 12) towards temperate BDT instead of boreal NET. In sampling units 13 and 16 DGVM failed to establish vegetation (predicting bare ground) where AR reported boreal BDS. RS represented the PFT profiles of the AR reference well in most cases but tended to overestimate the frequency of dominance by C3 grasses at several locations (plots 3, 16 and 20). While DM showed no general spatial pattern of PFT profile deviations from the reference dataset, PFT profiles of plots 4 and 19 obtained by DM had almost no similarity to the corresponding profiles of the AR reference dataset: C3 grasses and boreal BDT were predicted instead of bare ground and boreal NET, respectively.


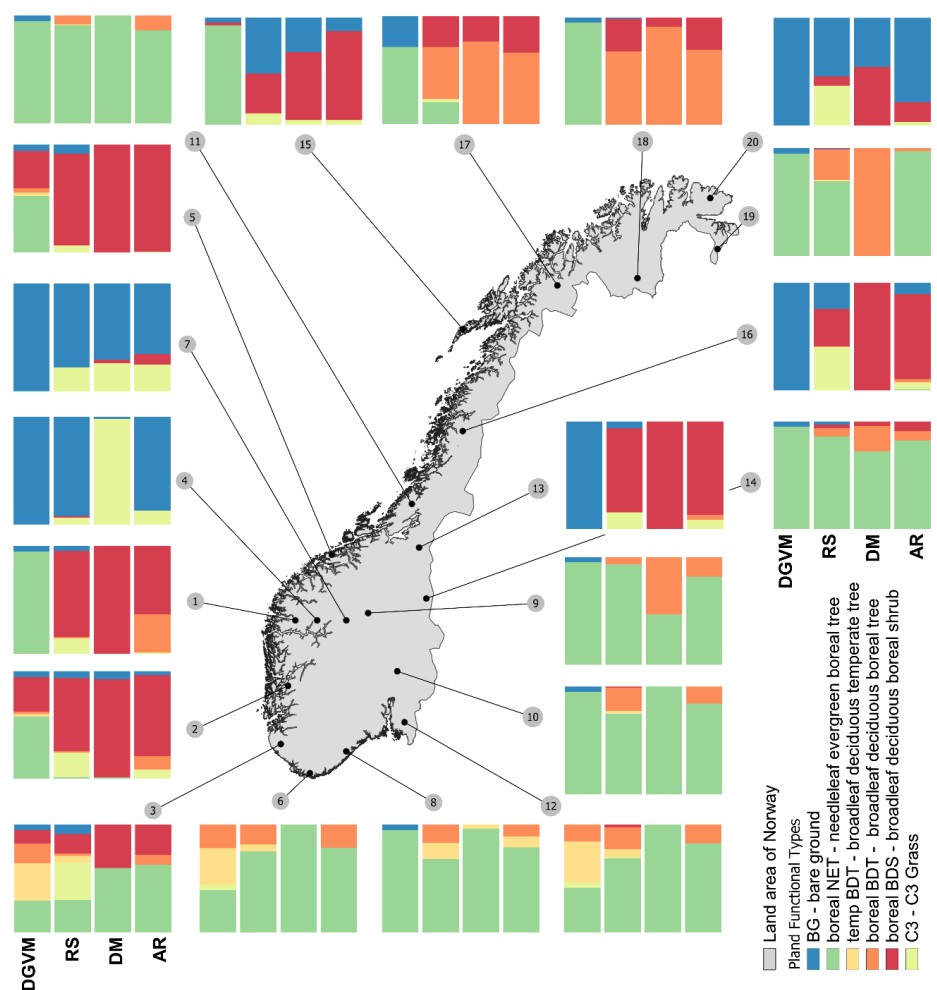

**Figure 3 – PFT profiles for each of the 20 plots for the three methods compared in this study and the AR reference dataset. The columns in each cluster of four bar-charts represent, from left to right, the methods dynamic global vegetation model (DGVM), remote sensing (RS) and distribution model (DM), with the AR reference dataset to the right.**

## 4    Sensitivity experiments and model improvement

### 4.1    Methods

We used the results of PFT profile comparisons between DGVM and the AR reference and the results obtained for the DM dataset as a starting point for exploring possible relationships between the poor performance of DGVM and DGVM parameter settings. We first identified the three most abundant PFTs (i.e. boreal NET, boreal BDT and boreal BDS) in our set of plots (Table S4). Thereafter, we identified the major VTs that were translated into these PFTs to be pine forest, birch forest and dwarf shrub heath, respectively (Table 4). We selected three of the most important environmental predictors for the distribution of each of these VTs, as identified by DMs (see Horvath et al. 2019) for sensitivity experiments of DGVM parameter settings (Table 4): snow water equivalent in



October (swe_10), minimum temperature in May (tmin_5) and precipitation seasonality (bioclim_15). We used
frequency-of-presence plots (i.e. graphs showing variation in the abundance of the VT as a function of an
environmental variable) to identify threshold values for presence of the three VTs and implemented these threshold
values into DGVM as new limits for establishment of the three PFTs as shown in Table 4 (also see Fig S11).
We explored the extent to which revised parameter settings improved the performance of DGVM on the subset of
six plots (i.e. numbers 1, 2, 5, 15, 17 and 18) in which the boreal NEB was most strongly overrepresented compared
to the AR reference dataset. Sensitivity experiments were carried out by a stepwise process, in each step adding
one new threshold, specific for the three PFTs at the same time. Parameters were added in the following order:
swe_10, tmin_5 and bioclim_15 (only relevant for the boreal NET). Only the results of DGVMs with all three
parameters changed are reported here (results of the other two experiments are summarised in Table S12). For
example, in the three sensitivity model runs (i–iii), (i) the requirement for establishment of boreal NET was set to
swe_10 > 150 mm; in (ii) and (iii) the additional demands tmin_5 > –5 °C and bioclim_15 < 50, respectively, were
enforced.
**Table 4 – New parameter thresholds for establishment of the three PFTs explored in DGVM sensitivity experiments.**
**Variables for which parameter settings were explored were: swe_10 – snow water equivalent in October given in mm;**
**tmin_5 – minimum temperature in May (°C); bioclim_15 – precipitation seasonality (unitless index representing annual**
**trends in precipitation).**

| VT | PFT | SWE_10 (mm) | Tmin_5 (°C) | Bioclim_15 |
|---|---|---|---|---|
| 2ef – Dwarf shrub heath / Alpine calluna heath | Boreal broadleaf deciduous shrub | > 380 | > -10 | – |
| 4a – Lichen and heather birch forest | Boreal broadleaf deciduous tree | > 180 | > -7.5 | – |
| 6a – Lichen and heather pine forest | Boreal needleleaf evergreen tree | > 150 | > -5 | < 50 |


### 4.2   Results

Adding new parameter thresholds in accordance with Table 4 made PFT profiles identified by DGVM more similar
to those of the AR reference dataset for four out of the six plots in the experimental subset (1, 2, 5 and 15): in plots
1 and 15, Boreal NET was correctly replaced by boreal BDS; in plots 2 and 5 boreal NET was replaced by boreal
BDT, BDS and temperate BDT. Addition of new parameter thresholds also reduced the modelled abundance of
boreal NET in plots 17 and 18, but DGVM failed to populate these plots with another PFT (Fig. 4). The improved
performance of DGVM on the experimental sampling units was mainly due to the implementation of the threshold
for bioclim_15, while the changes made for swe_10 and tmin_5 had little impact on the results (Table S12).
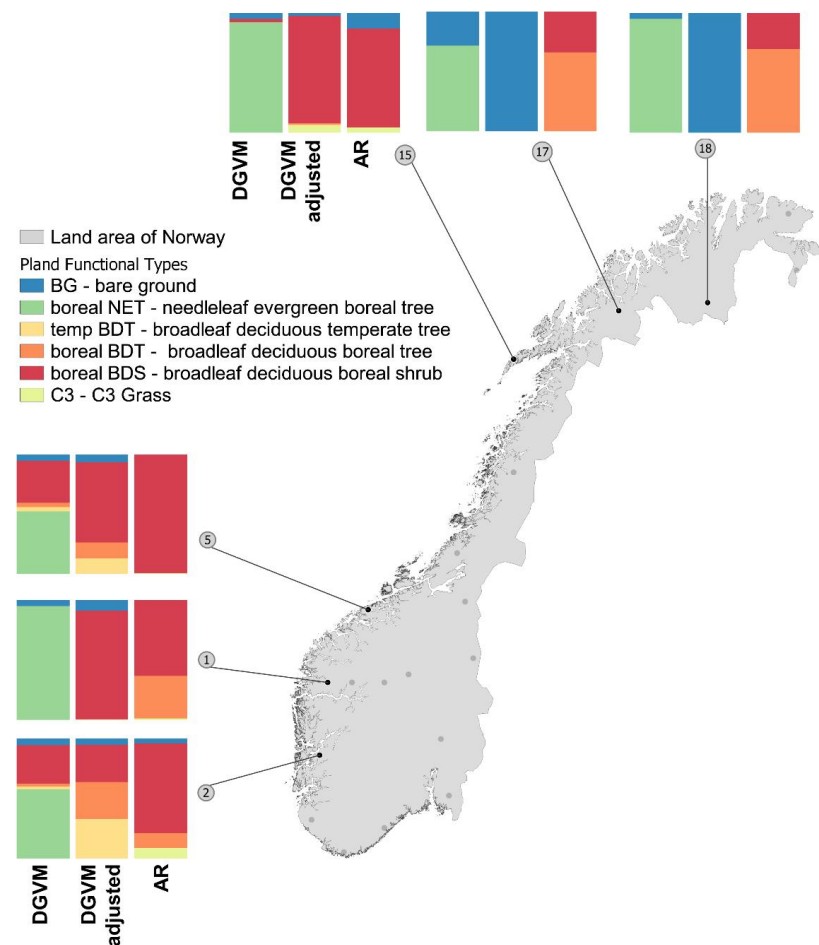


**Figure 4 – PFT profiles for the subset of six plots subjected to sensitivity experiments with new DGVM establishment thresholds. The columns in each cluster of three bar-charts represent, from left to right, dynamic global vegetation model (DGVM) with original (default) parameter settings, DGVM with revised parameter settings, and the AR reference dataset. For further details, see Table S12.**

**5    Discussion**
**5.1    Comparison of PFT profiles**
The maps of PFT distributions generated by DM and RS are generally similar (Fig. S8) across most of our study
area. This indicates that output from DM, which is rarely used for evaluating PFT distributions from DGVMs, can
be used for this purpose in addition to the commonly used RS-based datasets. There are, however, some differences
between results obtained by the two methods near the northern Norwegian coast and in the mountain areas of
western Norway which will be discussed below.
We recognise six possible explanations for the differences in PFT profiles obtained by DGVM, RS and DM for
the 20 plots (see Table 5), related to the following issues: (i) the conversion scheme (ref. Table 2); (ii) what is





actually modelled by DGVM, RS and DM, e.g. in terms of potential vs actual vegetation; (iii) the performance of
individual DM models; (iv) transforming predictions from single DMs into a seamless vegetation map, i.e. that
assigns one VT to each pixel; (v) DGVM performance; and (vi) missing PFTs in DGVM.

### 5.1.1   The conversion scheme

The conversion schemes used to reclassify vegetation and land cover classes into PFTs have been reported as a
possible attributor to erroneous PFT distributions (Hartley et al., 2017). While we use a simple conversion scheme
which assigns each land cover type/vegetation type to one and only one PFT (Dallmeyer et al., 2019), more
complex conversion schemes exist, by which each land cover class is translated into a multi-PFT composition that
co-occur within a grid cell (Bonan et al., 2002; Li et al., 2016; Poulter et al., 2011; Poulter et al., 2015). Our
approach may be advantageous when the classes to be converted are homogeneous, in the sense that one PFT is
clearly dominating in the type, and in the sense that the range of variation within the class in PFTs is negligible,
such as is the case for 90% of the DM- and RS-classes in our study. Our simple scheme may, on the other hand,
be a source of bias when quantitatively important VTs are ambiguous in one way or the other, or, more commonly,
in both ways at the same time. The set of VTs used in our study includes several relevant examples: VTs that may
include a wide spectre of tree-dominant types; the VT '*1a/1b - Moss snowbed / Sedge and grass snowbed*' which
covers a range of variation in the relative abundance of graminoids and, hence, shows affinity to C3 as well as to
BG; and the VT '*8a - Damp forest*', which is usually dominated by the evergreen Scots pine and converted into
boreal NET, but that in some instances (e.g. after clear-cutting) is dominated by deciduous trees like *Betula* spp.
and should then be converted into boreal BDT (Bryn et al., 2018). However, a close inspection of DM shows that
our method reproduces similar PFT profiles as the reference dataset for all plots except two out of 20 plots (the
two outliers on Fig. 2, represented by plots 4 and 19 in Fig. 3).
In our case, a more complicated conversion scheme is likely to be compensated for by the sub-grid complexity
introduced in the process by which PFT profiles are obtained. Rather than estimating a PFT profile for the 1-km$^2$
plot directly, i.e. in one operation as in DGVM, the RS-based classes and VTs are first converted into PFTs in their
original resolution, and then subsequently subjected to aggregation to obtain the PFT profiles. This results in a
sub-grid PFT heterogeneity that could otherwise be implemented by using a more complex conversion scheme.

### 5.1.2   What is modelled by DGVM, RS and DM

The methods used in this study produce different representations of the vegetated land surface in terms of actual
or potential natural vegetation (Table 5). In order to model future vegetation changes and feedbacks, functional
type-based models like DGVM implicitly address the processes that control the distribution of vegetation (Bonan
et al., 2003; Song et al., 2013). Simulating natural vegetation processes under a given climatic equilibrium scenario
(at any given time), DGVM produces a model of potential natural vegetation (ex. Bohn et al., 2000, Hengl et al.
2018). RS-based classifications, on the other hand, describe the land surface at a specific time-point or changes
through time (e.g. Arctic greening and browning) (Myers-Smith et al., 2020) and, accordingly, portrays actual
vegetation as influenced by previous and ongoing land use (Bryn et al., 2013). Depending on the modelling setup,
DM may pragmatically describe the current ecological envelope of a target or aim at revealing the proximate
causes for its distribution (Ferrier and Guisan, 2006), thus modelling either actual or potential natural vegetation,
depending on the input data used for modelling (Hemsing and Bryn, 2012; Hengl et al., 2018).





In this study, we carefully restricted our attention to PFTs that represent natural vegetation, excluding VTs with
strong anthropogenic influences. This was done for all methods and the AR reference. Nevertheless, differences
with respect to what is actually modelled by the different methods, potential vegetation by DGVM and actual
vegetation by RS and DM, may have contributed to the observed among-model differences in PFT profiles.

### 5.1.3   DM performance

While the performance of the DM method is overall good, two plots stand out by PFT profiles that deviate strongly
from the AR reference (Fig. 2). For plot 4, the discrepancy is due to VT "*1a/1b - Moss snowbed / Sedge and grass*
*snowbed"*, which is represented by one of the best performing among the 31 DMs. For this VT, conversion scheme
bias is a more likely reason for the deviant PFT profile. For plot 19, boreal BDT is modelled because the VT
predicted by DM is *"4a – Lichen and heather birch forest"*. The fact that the DM for this VT is among the inferior
DMs (see the ranking of individual models presented in Horvath et al. (2019)) makes this explanation more likely
in this case.

### 5.1.4   Transformation of single-DM predictions into a vegetation map

The performance of DM on the particular plots may also be influenced by the method chosen for transforming
predictions from one DM for each VT into a seamless vegetation map. Assigning to each grid cell the VT with the
highest predicted probability of presence in that cell, which is a commonly used method for this purpose (Ferrier
and Guisan, 2006), favours VTs represented by good DMs. This is brought about by good DMs having a
distribution of predictions that is more spread out (with larger predictions for the pixels identified as the most
favourable cells) than poor DMs (Halvorsen, 2012). Alternative methods for this purpose should be tested in the
context of DGVM evaluating.
To avoid uncertainties associated with conversion between type systems and perhaps even further improve the
performance of DM, we recommend exploring the option of using PFTs directly as targets in DM. Direct modelling
of PFTs rather than taking the detour via VT models may reduce the number of environment predictors required
(116 layers used in Horvath et al. (2019)) in addition to circumventing the complicated process of modelling
thematically narrow vegetation types (VTs). Another potential advantage of modelling PFT targets directly is that
the model parameters will then be PFT specific, and not in need of being converted (from VT into PFT).
To further reduce the biases and uncertainties of DM-based PFT profiles, we recommend exploring the use of
variables derived from RS directly as predictors in DM. Previous studies have shown that RS -based predictors
may enhance DM performance on different scales: on vegetation-type level (Álvarez-Martínez et al., 2018); on
the habitat-type level (Mücher et al., 2009); and on the PFT level (Assal et al., 2015). Further suggestions for
improvement of the methods used in this study are found in Table 5.





**Table 5 – A summary of the key properties of the three methods compared in this study. DGVM – dynamic global**
**vegetation model, RS – remote sensing, DM – distribution model, AR – reference dataset.**

| Key property | Method | | |
|---|---|---|---|
| | DGVM | RS | DM |
| Modelled property | Process-based vegetation model – using on *a priori* parameterizations | Classification based on satellite imagery (spectral reflectance) | Statistically based model of a target (response) and the environment (predictors) |
| Main purpose | Feeding vegetation changes into ESM for further quantification of feedbacks between land surface and the atmosphere | Mapping of land cover or land use for descriptive purposes, management or monitoring | Predicting the spatial distribution of a target and/or to summarise its relationship with the environment |
| Material | Climate forcing, PFT parameters, host model | Satellite imagery in different bands | Presence-absence training data, environmental predictors |
| Spatial extent | Global to regional (Single-cell tests) | Global to local | Regional to local |
| Modelling outcome | Potential vegetation | Actual vegetation | Potential or actual vegetation, depending on the training data |
| Advantages | – Addresses the processes<br>– Feedback loops with other Earth system components can be included<br>– Continuous temporal scale of prediction into the future | – Observation-based<br>– High spatial resolution<br>– Good temporal coverage | – Opens for use of proxies for important predictors<br>– May provide insight into drivers of distributions |
| Disadvantages | – Low performance (e.g. compared with RS and DM) as long as the underlying processes are not fully understood and properly parameterised<br>– Parameter intensive | – Data are sensitive to cloud cover and shaded areas<br>– Atmospheric correction needed<br>– Provides limited insight into the processes that regulate the distributions of land cover types<br>– No feedback included | – Provides limited insight to the processes that regulate the distributions of targets<br>– Temporally static (one time-point addressed by each model)<br>- No feedback included |
| Possible interactions with the other methods | – May improve DM by pointing at relevant predictor variables<br>– May improve RS by identifying threshold values | – May improve DGVM by improved parameterization (based on RS indices)<br>– May improve DM by providing predictor variables, directly or as indices (NDVI, PAR etc) | – May improve parameterization and envelope discrimination of DGVM<br>– May improve RS by targeting specific PFTs that have similar reflectance, but different ecology |


### 5.1.5    DGVM performance


Our results show that, for many plots, the PFT profiles simulated by DGVM differs from those of the reference
dataset. According to our results, DGVM overpredicts the coverage of bare ground and boreal NET and
underpredicts the cover of C3 grasses, boreal BDT and boreal BDS. While the AR reference dataset shows that
the northern plots (specifically plots 17 and 18) are covered by mountain birch forest and shrubs (boreal BDT and
boreal BDS), DGVM predicts dominance of boreal NET in these plots. Overestimation of boreal NET has also
been reported by Hickler et al. (2012) for large parts of Scandinavia, who attributed this to the lacking
representation of shade tolerance classes in DGVM models. A similar pattern is seen in our results: the PFT profiles
obtained by DGVM during the 400-year spin-up (Fig. S10) show no sign of boreal BDT in the early phases of
model prediction, as expected of an early successional forest in Norway.
The western parts of Scandinavia are dominated by shade intolerant birch forests (Bryn et al., 2018) which
gradually give way to coniferous forests along the oceanity-continentality gradient towards east (Wielgolaski,
2005). The overprediction of DGVM in the west indicates that the DGVM does not only lack shade-intolerant
PFTs, but also that improved representation of winter-time respiration loss and soil frost-induced drought stress of





boreal NET in spring in regions with higher temperature fluctuations around 0°C during winter time compared to
the more continental regions (see e.g. Oksanen, 1995; Sevanto et al., 2006) are needed.
Our results further suggest that the DGVM underrepresents grasses and shrubs compared to the reference dataset.
This may be explained by the built-in constraints in the light competition scheme of DGVM. For example Oleson
et al. (2013) mention that regardless of grass and shrub productivity, trees will cover up to 95% of the land unit
when their productivity permits. The priority given to a PFT in DGVM decreases with the stature of the organisms
in question because of the increasing probability that a lower layer is covered by another layer. The degree of
underrepresentation is therefore expected to increase from shrubs to grasses. Accordingly, DGVM predict
dominance by trees in the most productive regions, by grasses in less productive regions, and by shrubs in the least
productive non-desert regions (Zeng et al., 2008). The underrepresentation of C3 grasses by DGVM across the 20
study plots in our study accords with the results of Zhu et al. (2018), who found that C3 grasses are underpredicted
on a global level in an earlier version of DGVM.
Inappropriate parameterisation of shrubs may be a reason why the DGVM underestimates boreal BDS in many of
the coastal plots (1, 2, 5, 15) (Table S6). The implementation of shrubs as a new PFT in an earlier version of
DGVM (CLM3-DGVM) by Zeng et al. (2008), which is parameterised for representation of taller shrubs with
heights between 0.1 and 0.5 m, may not suit the majority of dwarf shrubs (of genera *Calluna, Betula, Empetrum*)
that abundantly occurs in Norwegian ecosystems. To this, Castillo et al. (2012) add that the sparse shrub and grass
vegetation cover simulated by DGVM in the tundra regions may be caused by the soil moisture bias inherited from
the host land model CLM4 (Lawrence et al., 2011). Another reason for DGVM's underestimation of boreal BDS
in coastal areas could be the 4000-yr tradition of coastal heath management in Norway (Bryn et al., 2010) which
causes a large discrepancy between the actual vegetation modelled by RS, DM and AR and the potential natural
vegetation simulated by DGVM under present-day climatic conditions (e.g.  Bohn et al., 2000, Hengl et al. 2018).
We therefore argue that more sensitivity studies of PFT-specific parameters for height, survival, establishment
etc., across all PFTs, are needed.
Despite the shortcomings discussed above, DGVM performs reasonably well for some PFTs. One example is the
temperate BDT, which is correctly predicted by the model to be restricted to the southern coastal plots (Bohn et
al., 2000; Moen, 1999). This finding suggests that some climatically driven PFTs (i.e. temperate BDT) are well
implemented by the existing parameters in the current DGVM.

### 5.1.6    Missing PFTs

DGVM coerces the World's immense variation in plant species composition (vegetation) into a very limited
number of predefined PFTs, compared to classification schemes used by the other methods in this study (RS, DM
and AR; see Table 2) and by other approaches to systematisation of ecodiversity (e.g. Dinerstein et al., 2017; Keith
et al., 2020). In particular, the number of high-latitude specific PFTs is insufficient to realistically represent the
biodiversity of these ecoregions, as pointed out by Bjordal (2018) and Vowles & Björk (2017). Comparisons
between PFT profiles obtained by DGVM and profiles obtained by DM may suggest specific vegetation types that
need to be better represented in DGVMs, either by improving an existing PFT or by adding a new PFT (e.g. dwarf
shrub vs. tall shrub; moss dominated snow-beds, wetlands, lichens). In our study, the PFT profile of DGVM is
represented by the six boreal PFTs, whereas the original data for RS, DM and AR include an average of 17% (ref.
Table S4) of the total area which cannot be represented by these six PFTs (classes for "Excluded" PFT category

 

ref. Table 2). This reminds us of the missing PFTs in the classification scheme of the DGVM, but it also points to
the problem that certain ecosystems in our study area do not have a real representation in the PFT schemes of
DGVM. This is exemplified by wetlands; important ecosystems that are still not represented in many of the current
DGVMs. This is not only problematic from the perspective of land surface energy balance (Wullschleger et al.,
2014), but also brings issues of carbon storage and cycling, and other interactions between the land surface and
the atmosphere (Bjordal, 2018).
Our results demonstrate a great potential for increasing the thematic resolution of DGVMs in terms of developing
and parameterizing new specific PFTs to be representative of the high-latitude and high-altitude habitats, as
exemplified by Druel et al. (2017) and also deriving parameters from observations, DMs or RS products (Bjordal,
2018; Wullschleger et al., 2014), specific for the high latitudes (Druel et al., 2017).

**5.2    Sensitivity tests**

Adjusting DGVM parameters so that they correspond better with environmental drivers known to be functional in
the high-latitude PFTs has been suggested as a measure to improve the performance of DGVM in these parts of
the World (Wullschleger et al., 2014). Our simple sensitivity experiments demonstrate that DM results can inform
parameterisation, in DGVM, of the range along variables used in DM where a PFT occurs. Most notably, we
recognized three important environmental drivers for the distribution of high-latitude PFTs not yet represented
well in DGVM. This adds to environmental thresholds for establishment, survival or mortality of a PFT previously
used in DGVMs to restrict the predicted distribution of PFTs to realistic geographic regions (Miller and Smith,
471 2012).
Adjustment of the climatic thresholds for the establishment of the high-latitude PFTs (i.e. boreal NET, BDT, BDS)
seemingly bring the PFT profiles of DGVM closer to those of the reference data (Fig. 4). In particular, the
sensitivity experiments with DGVM highlight the importance of precipitation seasonality (i.e. bioclim_15) as a
critical limiting factor for the establishment of boreal NET. While some studies have emphasized the importance
of seasonal distribution of rainfall on vegetation in the semi-arid areas (Zhang et al., 2018), the importance of this
factor for high-altitude areas is less well studied (Oksanen, 1995; Sevanto et al., 2006). Better representation of
the processes related to the response of boreal NET to water availability, especially spring-drought in DGVM, also
warrants further investigation. From our results for Sites 17 and 18, we notice that adjusting the climatic thresholds
for growth of boreal NET does not automatically make other PFTs grow. Boreal BDT and BDS can establish at
both sites, but their growth rates are too slow to make them occupy a large area at these sites. This prevents
development of similarity with the PFT profiles of AR reference dataset (Fig. 4) and implies that other
environmental conditions, e.g., nitrogen availability, might play a more important role in limiting the growth of
BDT and BDS in CLM4.5BGCDV. The biases of DGVMs in simulating boreal broadleaf deciduous tree and shrub
has been widely noticed in other studies (Castillo et al., 2012), and should be investigated further.
While going into further details of which additional PFTs should be included in DGVMs and how these and other
PFTs should be parameterised is beyond the scope of the present paper, we emphasize the potential of using DM
for improving the parameters of DGVMs. More specifically, we propose more intensive exploration of DM as a
tool for identification of potential environmental drivers for the high-latitude PFTs, which may enhance the
performance of DGVMs in high-latitude ecoregions.





## 6 Conclusions

This study emphasizes the potential of using distribution models (DM) for representing present-day vegetation in evaluations of plant functional type (PFT) distributions simulated by dynamic global vegetation models (DGVMs) and for improvement of specific PFT parameters within DGVMs. By identification of the main differences among PFT profiles obtained by three methods (DGVM, RS and DM) in selected high-latitude plots distributed across climatic gradients in Norway, we show that PFT profiles derived from DM and RS are in the same range of reliability, judged by resemblance to a reference dataset (AR). Hence, we suggest that DM results can be used as a complementary evaluation dataset to benchmark the present-day DGVMs. This approach is recommended when high-quality RS products are not available.

Comparing the twenty PFT profiles obtained by DGVM with those obtained by AR shows a large overestimation by DGVM of boreal needleleaf evergreen trees (boreal NET) and bare ground at the expense of boreal broadleaf deciduous trees and shrubs. This is attributed to missing processes and PFT parameterizations of high-latitude PFTs in DGVM. We use DM results to identify three new PFT-specific environmental parameters which, in a series of sensitivity experiments, improve the distribution of boreal NET predicted by DGVM. The new PFT-specific thresholds for establishment decrease the bias of boreal NET in DGVM across four out of six plots. We argue that these new thresholds should be transferable to other DGVMs simulating high-latitude PFTs, and that our DM-based approach can be transferred to other ecosystems.

Further development of DGVM, such as refining parameters for existing boreal PFTs and increasing the thematic resolution of PFTs for boreal areas, should be strongly encouraged to achieve a more realistic simulation of the distribution of actual vegetation by DGVM, to increase the reliability of future predictions, and the reliability of predicted vegetation feedbacks in the climate system.

## 7 Acknowledgements

NIBIO is acknowledged for providing access to the area-frame survey AR18X18 dataset. UNINET Sigma2 is acknowledged for providing computing facilities. Geir-Harald Strand is acknowledged for providing scientific assistance and Michal Torma for providing technical assistance.

## 8 Data availability.

The scripts used in this study are available in the GitHub repository https://github.com/geco-nhm/DGVM_RS_DM_Norway. High-resolution DM-based and RS-based PFT maps are available from the authors on request (Fig. S8). DGVM outputs are provided in the Table S9, Table S12 and Fig. S10.

## 9 Author contributions.

All authors have contributed to conceptualizing the research idea. PH curated the data and was responsible for the distribution modelling and for compiling and analysing the data from all methods. HT carried out the modelling and sensitivity tests using the DGVM (CLM4.5-BGCDV). PH together with AB and RH were responsible for writing, with all authors contributing to reviewing and editing the paper. FS, AB, TKB and LMT acquired funding for this research.



**10   Competing interests.**
The authors declare that they have no conflict of interest.
**11   Financial support.**
This work forms a contribution to LATICE (https://www.mn.uio.no/latice), which is a Strategic Research Initiative
funded by the Faculty of Mathematics and Natural Sciences at the University of Oslo (UiO/GEO103920). It is also
part of the EMERALD project (294948) funded by the Research Council of Norway.

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
