# Peer review of "Improving the representation of high-latitude vegetation distribution in Dynamic Global Vegetation Models"

_Biogeosciences, 2020_

## Referee Comment (RC1) · Anonymous Referee #1 · 30 Jun 2020

General Comments

The overall objective of this paper was to identify biases in a dynamic global vegetation model (DGVM) and, if possible, to find ways of reducing the biases. The analysis focused primarily on relatively undisturbed landscapes in Norway. The target model output was the within-gridcell plant functional type (PFT) distribution. One unique and valuable aspect of the manuscript was that the PFT distributions predicted by the DGVM were compared to multiple products, including field surveys, satellite products, and the output of species distribution models. Field surveys were much more similar to the satellite products and distribution models than to the DGVM. Improvement to the

[Figure]

DGVM was realized by incorporation of a precipitation seasonality index, although it was clear that this improvement would not be the end of the story.

Given that PFT distribution is an important quantity that is still challenging for DGVMs to predict, I think that the manuscript covers a topic that will be interesting and useful to readers of Biogeosciences. I also appreciated how the DGVM was compared to multiple products and how the distribution model was leveraged. However, I think that the value of the manuscript could be increased by being more thorough with the methods (see below). Also, I think that more could be done to make the manuscript interesting to readers who use models other than CLM.

Specific comments

The title should be modified. It mentions "Dynamic Global Vegetation Models" in the plural, but only one model is discussed. I also think the title is too general. I would suggestion "high-latitude vegetation distributions" rather than simply "high-latitude vegetation".

Lines 83-84: This point is overstated. There are publications that have evaluated PFT distributions from dynamic vegetation models against field-based datasets, at least on regional and national scales.

Methods: I am puzzled by the limitation of the study to only 20 plots. Certainly these 20 plots span the range of mean annual temperature and precipitation, but other factors are also commonly perceived to be important. Indeed, the distribution model seems to take 100+ inputs. Some questions that come to mind is whether the plots span the range of observed precipitation seasonality (identified by this study as an important factor!), soil texture, and soil nutrients.

Line 157: Why not assign the observed soil texture to the 20 plots?

Section 2.4.3: I am concerned that the DGVM and the DM uses different driver data to represent the same phenomenon. For example, does one use SeNorge2 and the

other reanalysis to represent precipitation? Does one use observed soil texture and the other "default" soil texture? If so, might differences in inputs account for differences in the DGVM and DM predictions?

Line 183: Was the DM model previously tuned to these 20 plots? To Norway?

Line 414: Might phenology also be an issue? Further, what is the light compensation point of the PFTs? Perhaps the authors can use the light compensation point to directly evaluate the relative shade tolerance of the different PFTs.

Discussion: Are there lessons for people who use other models? The more the authors can draw out such lessons, the broader the audience this paper would appeal to. The TEM model, which has a more detailed representation of boreal PFT diversity than CLM, immediately comes to mind as one example.

Technical corrections

The manuscript is very readable, but it should still be reviewed for grammar.

Page 3, Lines 43-45: There is a problem with word choice in this sentence. Vegetation distributions are not implemented in ESMs, but rather are predicted by ESMs. The ESM predictions can then be evaluated with satellite products (as done in the present analysis).

Section 2.4.1: It would be useful for the authors to briefly describe how the DGVM determines the amount of area to each PFT.

Data availability: Note that the GitHub link not up yet. I understand if the authors do not want to release the link prior to manuscript acceptance, but it is still important not to forget to release the link.
* * *

---

## Referee Comment (RC2) · Anonymous Referee #2 · 19 Aug 2020

General Comments

This study evaluates estimates of PFT distributions from a DGVM in comparison to those of remote sensing and empirical models, and against a field-based dataset, for 20 plots of high-latitude vegetation types across Norway. The topic investigated, approach taken, and results reported will be of interest to the modeling community.

The paper could benefit from more or better explanation of the methods, especially the CLM simulations. For example, it is unclear whether or not this is intended to be any kind of 'temporally-explicit' analysis; this seems a sort of model estimation of some 'average' PFT distribution from the spin-up results that was compared to field plots and

remote sensing data, both of which presumably represent a specific point in time (that is not specified in either case in the methods here).

To properly interpret the results, the sensitivity tests need more explanation and clarification to justify and understand what was done here in this study (vs. previous work).

The "RS method" as one of the three methods compared here seems kind of out of place in this analysis since it is not a method for predicting future PFT distributions as with the DGVM and DM methods. What is the reasoning / purpose behind including RS in this comparison? Or could / should it be used in this study more as a 'reference' data set, like the AR data?

Specific Comments

25-26. please consider this statement carefully; numerous authors could claim that this is untrue

34. can these three thresholds be named here, or at least hint at what they are (e.g. "... based on ...)?

115-116. this is not quite clear and perhaps needs to be specified or qualified; i.e. don't many "countries" have national-scale inventory programs?

126-131. Selecting only 20 plots seems limited, even if deemed acceptable for bioclimatic variation. There needs to be better explanation / justification for this choice, how "acceptable" was determined, and whether a kriging of temperature and precipitation really captures "bioclimatic" variation across the country.

150. curious decision to give a new acronym to CLM. why not just refer to it as "CLM"? and actually, you do, somewhat, as it seems to switch back-and-forth between "DGVM" and "CLM4.5" for the rest of the manuscript. I see the idea to associate the results from CLM as representative of the "DGVM" approach, but when describing or referring to the specifics of CLM then just call it "CLM" (or "CLM4.5")

[Figure]

154. it may be useful here to point out what these simple assumptions are, and how different (or not) they are from those for which the DM method is based on.

171. was soil C initialized somehow, or was it a separate (longer) spin-up? are these mostly undisturbed sites or was that taken into consideration for the vegetation spin-up at each site? was the CORDEX climate used for the spin-up? average or de-trended?

174. what year / era does this RS map represent? Table 2. I don't think all of this detail is necessary in the main text.

278, 279 & 305 are confusing uses of sub-headings

287. swe_10 and tmin_5 make sense as described but can "precipitation seasonality" be explained? "bioclim_15" is not as obvious as the other two parameters

293-299. there just seems like so much of the justification and explanation of decisions and approaches for the sensitivity test are glossed over here. For example, why are these particular parameters chosen, how was bioclim calculated, is the stepwise order important, what does it mean "three PFTs at the same time", how were the thresholds determined, etc etc. Perhaps a little more explanation than just "see Horvath et al 2019" (line 286) would be helpful.

414-415. this seems like a bit of a leap without a more direct connection to the results of this study.

468. but in line 312 it was stated that two of those three "had little effect"

498-499. when are high-quality RS products ever not available anymore in this day-and-age?

503. Just to be clear, it seems that these parameters were identified in a previous study, not this one, correct? And actually in this study only one of them (bioclim_15) was found to be useful, no? This same claim is made in the abstract, as well, and should be used with care.

Technical Corrections

- please review the grammar, wording and sentence structure throughout

42. please re-word and fix the grammar of this sentence one way or the other

55. remove "the" before DGVMs

60. latitudes

150. replace "further" with "hereafter"

170. "recalculated"

Table 2. "AR" is missing from the caption

292. change "NEB" to "NET", I think

341. "spectre" should be "spectrum"?

412. "overprediction of Boreal NET"?

---

## Referee Comment (RC3) · Anonymous Referee #3 · 25 Aug 2020

The manuscript "Improving the representation of high-latitude vegetation in Dynamic Global Vegetation Models" by Horvath et al analyses the performance of three different vegetation modeling approaches with regard to the spatial distribution and relative abundance of plant functional types (PFT) in Norway. The modeling approaches include a dynamic global vegetation model (DGVM), remote sensing (RM), and a statistical distribution model (DM), which relates occurrences of vegetation types to multiple environmental variables. The authors found that both RM and DM showed a better performance than the DGVM when compared to observational data from a range of field sites. They then tested if it was possible to use the DM to improve the predictions of the DGVM with regard to PFT composition and distribution. It was found that,

through inclusion of three further bioclimatic constraints based on the analysis of the DM, the performance of the DGVM could be improved. The authors recommend DM as a complementary tool for the assessment and improvement of DGVMs.

The manuscript is well written and easy to understand in general. The research topic (assessing and improving DGVMs at high latitudes) is certainly relevant, and the chosen approach is original and seems useful to me. However, the description of the methods needs to be improved, with regard to the chosen statistical approaches, and also the motivation to carry out certain analyses. It often becomes clear only later in the manuscript why a certain method was applied. I therefore recommend minor revisions before a new version of the manuscript may be submitted.

Comments:

L 28 While the term 'DGVM' is explained at the beginning of the abstract, the term 'distribution model (DM)' is used in this sentence without previous explanation. Please explain shortly in the abstract what a DM is and how it differs from a DGVM, since some readers may not be familiar with the concept.

L 58 Please define or explain in more detail what you mean by 'thematic resolution'. Furthermore, it should be mentioned that recently, specific high-latitude PFTs, such as mosses, for instance, have been added to a number of DGVMs, e.g. Jules (Chadburn et al, 2015, The Cryosphere), JSBACH (Porada et al 2016, The Cryosphere), or ORCHIDEE (Druel et al 2017, Geoscientific Model Development) and several more.

L 60 Three examples are given for the difficulties of DGVMs to simulate extents of high-latitude PFTs correctly. However, I do not see how the underestimation of forest carbon storage by DGVMs relates to this, since this is rather a consequence, and not a reason for the incorrectly predicted extent. Please explain in more detail.

L 71 Please add a short statement to describe in which regard the RS products are not consistent.

[Figure]

L 83 At least one study (Druel et al 2017, Geoscientific Model Development), uses site data to assess the DGVM's performance with regard to plant traits. Please be more specific in this regard, and explain what exactly is new in the validation method.

L 121 I do not understand this sentence: If one plot is 0.9 km2 large, then 1081 plots are around 1000 km2, but 18x18 km are only 324 km2. Also, the plots are distributed throughout Norway, so the 18x18 km area has to mean something else. Is it the distance between the plots on a grid which covers Norway? Please explain.

L 129 To me it seems that low values of temperature and precipitation are underrepresented in the 20 selected plots compared to the full data set. This should be mentioned here briefly and then considered later in the Discussion section.

L 156ff By using the default surface parameter values for CLM, the DGVM may miss some relevant information to correctly predict PFT distribution, compared to RS and DM. Furthermore, by using climate forcing from 1980-2010 and running the DGVM into a steady state with regard to this period, historical climatic effects, which may influence today's PFT distribution are not considered. These points should be mentioned in the Discussion section of the manuscript.

L 162 Why was the CORDEX data not also used for the DM method? This should be briefly mentioned here.

L 175 Please explain 'supervised' and 'unsupervised' in more detail.

L 182 the number of explanatory variables (116) is rather high. It should be shortly explained what these are, and why such a large number is necessary for the regression. Even if this information is provided in Horvath et al (2019), it should be summarized here.

L 183 It would be good to add a short summary of the evaluation method for the DM here, so the reader can assess the DM better.

L 186 I wonder if, by discarding all other VT except the most probable one, biases in the

distribution of the VTs are introduced. Let us assume the logistic regression predicts a certain VT always with a slightly higher probability than a second one; according to the description, only the first VT would occur in the predicted map at all pixels, and all observations of the second one would be discarded, although this VT occurs quite frequently in reality. Please explain this in more detail.

L 200 I don't understand why an aggregated PFT profile is needed, I thought that the comparison of the 3 modeling approaches and the AR data is done for each of the 20 plots?

L 208ff This sounds like one comparison was done with the aggregated profiles (one for each method, aggregated over all 20 plots), using the chi-square test. Then, for each of the 20 plots the profiles were compared regarding their dissimilarity. It is not clear to me, why two different statistical methods were used to compare the models (DM,RS, DGVM) to AR.

L 222 I thought the dissimilarity index was used to assess the similarity between the 3 modeling approaches and the AR data. Why is it then necessary to do a pairwise Wilcoxon-Mann-Whitney test in addition? Please explain the reasons for the chosen statistical approach in a more detailed way.

L 230ff As mentioned above (L200), by aggregating the PFT profiles of the 20 plots, differences in profiles between plots are lost. Hence, it is not possible to evaluate the 3 models with respect to the correct prediction of differences in profiles between individual plots. Also, while the AR data (for each plot) can be interpreted as a random sample, it is not clear to me how the model approaches can be consistently included in this Chi-square test. Moreover, the number of elements (6 PFTs) is actually too small for a Chi-square test. The authors need to justify this better, or change their testing approach.

L 249 If I understand Fig. 2 correctly, the lines which connect the dots denote the individual plots, which means that for one method (e.g. DGVM), the dissimilarity can

be high (1.0), while for another method (e.g. RS) it can be much lower. The result that the goodness of the fit between a given method and AR data depends on the set of chosen plots may point to some underlying systematic deficiencies of each method and should be discussed later.

L 252 The statement in this sentence is not evident to me in Fig. 3, because this figure simply shows the profiles for each plot (which is a good way of illustrating the results, in my opinion). Wrong reference?

L 254 Please see also my comment to L 222; I assume that the authors use the Wilcoxon test to assess if the median values of the dissimilarity indices for the 3 models are significantly different from each other. However, I think it is more relevant how the models differ to each other with regard to the AR data. This information is contained in the values of the dissimilarity index, and it should be reported more clearly here. The pairwise comparison of the 3 models seems to me of secondary importance to assess the goodness of the fit to AR data.

L 262ff The visual comparison of the 3 models in Fig.3 and the associated description is more helpful to assess the modeling approaches than the statistical methods described before.

L 279ff This belongs into the Methods section. Explaining the sensitivity analysis earlier also makes it much easier to understand the goal of the overall approach.

L 287 The term 'precipitation seasonality' should be better described, in particular since it is found later that it is important to improve DGVM parameterization.

L 379ff The point about 'good' and 'poor' DMs is not clear to me. Why should poor DMs be used at all? Please explain, and also consider my comment above (L 186).

L 411 It may not be clear to readers why the lack of a shade-intolerant birch-PFT in the DGVM leads to the over-representation of NET in plots 17 and 18. The birch-PFT should rather have an advantage in mountainous regions compared to NET, which is

currently lacking in DGVMs. Please clarify.

L 450 Please check the literature for the recent progress in including high-latitude vegetation types into the PFT scheme of DGMVs, and add this to the discussion.

L 467 This sentence is hard to understand, please reformulate.

L 475 It should be mentioned if increased seasonality promotes or impedes growth of NET.

Supplement:

L 40 missing reference L 51 missing reference L 52 missing reference

L 55 The PFTs for this study are not in bold font, but shaded grey, please make this consistent.

L 56 The caption of Tab. S6 should be a bit more detailed: Is zbot the bottom height of the canopy (11.5 m above ground)? How is the coefficient of variation in precipitation seasonality computed?

L 90 The cover fractions in plots 801,2108,4268 are clearly not in a steady state. Please check if this significantly affects the results (e.g. by extrapolating the trends in cover), and repeat the DGVM runs, if necessary.

L 122 missing reference

Comments on style:

L 42 I think 'an' is not needed here.

L 55 'DGVMs' instead of 'the DGVMs'

L 60 'at high latitudes' instead of 'in the high latitude'

L 66 'in' not necessary

L 138 the second "of the" is not necessary

L 373 add 'the' before 'reason'

L 401 'differ' instead of 'differs'

---

## Author Comment (AC1) · 14 Sep 2020

*Dear Referees,*

*On behalf of all the co-authors I thank you for the insightful and constructive comments directed to the manuscript "Improving the representation of high-latitude vegetation in Dynamic Global Vegetation Models". We have prepared point-by-point responses to each of the comments and believe that further implementation of these in the revision will improve the quality of our manuscript. For convenience and reference, we have numbered the Referee comments with "RC-x.x", where the first "x" corresponds to the referee number and the second "x" to the respective comment. Each of our responses is offered below the respective comment emphasized in blue italics.*

*Kind Regards,*

*Peter Horvath*

**Contents**

**1 Anonymous Referee #1**

**General comments**

The overall objective of this paper was to identify biases in a dynamic global vegetation model (DGVM) and, if possible, to find ways of reducing the biases. The analysis focused primarily on relatively undisturbed landscapes in Norway. The target model output was the within-gridcell plant functional type (PFT) distribution. One unique and valuable aspect of the manuscript was that the PFT distributions predicted by the DGVM were compared to multiple products, including field surveys, satellite products, and the output of species distribution models. Field surveys were much more similar to the satellite products and distribution models than to the DGVM. Improvement to the DGVM was realized by incorporation of a precipitation seasonality index, although it was clear that this improvement would not be the end of the story.

Given that PFT distribution is an important quantity that is still challenging for DGVMs to predict, I think that the manuscript covers a topic that will be interesting and useful to readers of Biogeosciences. I also appreciated how the DGVM was compared to multiple products and how the distribution model was leveraged. However, I think that the value of the manuscript could be increased by being more thorough with the methods (see below). Also, I think that more could be done to make the manuscript interesting to readers who use models other than CLM.

*We are thankful to the Referee #1 for his/her positive response and constructive comments.*

**Specific comments**

RC-1.1 - The title should be modified. It mentions "Dynamic Global Vegetation Models" in the plural, but only one model is discussed. I also think the title is too general. I would suggestion "high-latitude vegetation distributions" rather than simply "high-latitude vegetation".

*This is a good suggestion. We shall adjust the title to specify that high-latitude vegetation distributions are considered. With regard to the plural mention of DGVMs, we believe that even though we tested this particular exercise only on one DGVM (namely CLM4.5BGCDV), the procedures/methods of implementing variables from DM as new parameters in DGVM can be used in multiple DGVMs not just the tested one (thus the plural form).*

RC-1.2 - Lines 83-84: This point is overstated. There are publications that have evaluated PFT distributions from dynamic vegetation models against field-based datasets, at least on regional and national scales.

*In line with response to Referee #3 on this same point (see also comment RC-3.6), we will adjust the formulation of the sentence and add a reference.*

RC-1.3 - Methods: I am puzzled by the limitation of the study to only 20 plots. Certainly these 20 plots span the range of mean annual temperature and precipitation, but other factors are also commonly perceived to be important. Indeed, the distribution model seems to take 100+ inputs. Some questions that come to mind is whether the plots span the range of observed precipitation seasonality (identified by this study as an important factor!), soil texture, and soil nutrients.

*We agree that a higher number of plots would have been beneficial. Ideally, we would want 1000+ plots or perhaps a regional/global simulation. However, labor-demanding preparation of all data layers for each*

*plot was one of the critical factors for this study and we had to find a compromise between what was practically possible and what was considered robust in terms of the aim of the study. From a methodological perspective, our opinion is clearly that a representative sample of 20 plots is sufficient to demonstrate the differences between the three methods of representing the vegetation distribution.*

*The gradients of precipitation and temperature are known to be among the most influential for vegetation distribution (e.g., Ahti et al. 1968; Bakkestuen et al. 2008), thus we have chosen to include these particular two variables when selecting the 20 plots. However, we also agree with the Referee #1 in the argument that the 20 plots' representativity across the range of precipitation seasonality should be tested (since this is identified as an important factor). We will therefore include a comparable test and add a third diagram to the Supplementary Figure S3. Please also see the response RC-2.6 to Referee #2 with a similar request.*

RC-1.4 - Line 157: Why not assign the observed soil texture to the 20 plots?

*The observed data on the 20 plots unfortunately do not include information about soil texture. The plots were mapped using wall-to-wall vegetation mapping, where only data about the type of vegetation cover are available.*

RC-1.5 - Section 2.4.3: I am concerned that the DGVM and the DM uses different driver data to represent the same phenomenon. For example, does one use SeNorge2 and the other reanalysis to represent precipitation? Does one use observed soil texture and the other "default" soil texture? If so, might differences in inputs account for differences in the DGVM and DM predictions?

*Absolutely. Ideally, we would use the same climate input data for both DM and DGVM. However, there are technical obstacles: DM uses multi-year monthly averaged climate data as input, while DGVM requires 3-hourly meteorological data as the input. SeNorge2 dataset, which is used in DM, has only daily data available, therefore can only be used for DM but not for driving DGVM. For DGVM, we had to use available reanalysis or regional climate model data for present day climate (CORDEX data in this manuscript). To compare the differences between the driving data for DGVM and DM, we have listed mean annual temperature and precipitation for both datasets in the table S1 and figure S5 of the supplement. There are indeed some minor differences between the two sets of driving data, however it is beyond this study to quantify the effect of these differences. We will devote a paragraph to clarify the potential bias this may imply in the discussion.*

*Soil texture does not come in as an explanatory variable in the DM, whereas DGVM is using soil texture as an important parameter affecting various processes in soil, such as soil temperature, moisture and organic matter decomposition. We will add a comment on the differences between the input data in the paper and discuss its potential implications.*

RC-1.6 - Line 183: Was the DM model previously tuned to these 20 plots? To Norway?

*The DM was not tuned specifically to these 20 plots. The training data for DM included the whole set of 1081 plots (across Norway) at a different thematic resolution (detailed vegetation types instead of PFTs) and at a scale of one point per polygon. Although the 20 plots were included as a subset of the total 1081 plots, we believe the influence is minimal, since they have gone through a spatial and thematic conversion. Moreover, the DM was evaluated with a completely independent dataset.*

RC-1.7 - Line 414: Might phenology also be an issue? Further, what is the light compensation point of the PFTs? Perhaps the authors can use the light compensation point to directly evaluate the relative shade tolerance of the different PFTs.

*Please also see comments to Referee #2 (RC-2.14) and Referee #3 (RC-3.26) regarding this paragraph in the discussion. Phenology is likely to be an issue, as evergreen plants seem to have advantage in competing with deciduous plants in general in the high-latitude region in the model. It is therefore suggested that stress for evergreen plants in winter and spring may not be well represented in the model to limit the growth of boreal NET in some regions. However, we admit that this issue is not well documented through our results and therefore have decided to remove this paragraph from the discussion.*

RC-1.8 - Discussion: Are there lessons for people who use other models? The more the authors can draw out such lessons, the broader the audience this paper would appeal to. The TEM model, which has a more detailed representation of boreal PFT diversity than CLM, immediately comes to mind as one example.

*Thanks for the suggestions. The present-day vegetation distribution outputs from dynamical vegetation models could more often evaluated by use of multiple products complementing the RS, such as by including DM and AR as presented in this study. We also believe that the procedure of identifying new parameter values from DM, running a set of sensitivity tests and implementing the sensible new parameters into a DGVM is not limited to CLM4.5BGCDV (the DGVM tested here) but transferrable also to other DGVMs, such as the TEM model. We will make sure that this is stated more clearly and include more thorough discussions with regard to applicability to other models in the revised manuscript.*

**Technical corrections**

RC-1.9 - The manuscript is very readable, but it should still be reviewed for grammar.

*We will carefully search the manuscript for grammatical errors.*

RC-1.10 - Page 3, Lines 43-45: There is a problem with word choice in this sentence. Vegetation distributions are not implemented in ESMs, but rather are predicted by ESMs. The ESM predictions can then be evaluated with satellite products (as done in the present analysis).

*We will rewrite the sentence according to the referee's comment.*

RC-1.11 - Section 2.4.1: It would be useful for the authors to briefly describe how the DGVM determines the amount of area to each PFT.

*We will add a brief description on how the area of each PFT (i.e. percentage cover fraction %) is determined by DGVM in the revised manuscript. The percentage cover fraction of each PFT is equal to the average individual's fraction projective cover ($FPC_{ind}$) multiplied by the number of individuals ($N_{ind}$) and average individual's crown area ($CROWN_{ind}$). $FPC_{ind}$ is a function of the maximum leaf carbon achieved in a year, while $CROWN_{ind}$ is related to dead stem carbon simulated by the model. $N_{ind}$ is mainly determined by establishment and survival rate controlled by establishment and survival threshold conditions.*

RC-1.12 - Data availability: Note that the GitHub link not up yet. I understand if the authors do not want to release the link prior to manuscript acceptance, but it is still important not to forget to release the link.

*This is an important point. We shall keep in mind that the scripts are to be made available as soon as the manuscript is accepted.*

**2 Anonymous Referee #2**

**General Comments**

This study evaluates estimates of PFT distributions from a DGVM in comparison to those of remote sensing and empirical models, and against a field-based dataset, for 20 plots of high-latitude vegetation types across Norway. The topic investigated, approach taken, and results reported will be of interest to the modeling community. The paper could benefit from more or better explanation of the methods, especially the CLM simulations. For example, it is unclear whether or not this is intended to be any kind of 'temporally-explicit' analysis; this seems a sort of model estimation of some 'average' PFT distribution from the spin-up results that was compared to field plots and remote sensing data, both of which presumably represent a specific point in time (that is not specified in either case in the methods here).

*Thank you for this to-the-point comment. We agree that more careful explanation of some aspects of the methods is necessary. We will adjust the manuscript with regard to the specific comments you provide here.*

*This study represents a temporally explicit analysis of the 'present-day' vegetation distribution. We agree that this needs to be emphasized more clearly. In line with further replies to RC-2.10, the temporal context will be specified for each of the three modelling methods as well as for the AR in the respective sub-chapter (2.4).*

RC-2.1 - To properly interpret the results, the sensitivity tests need more explanation and clarification to justify and understand what was done here in this study (vs. previous work).

*We will add more explanation and discuss on the sensitivity tests in the revised manuscript. Also, we shall review the formulations of what was done in this study vs previous work.*

RC-2.2 - The "RS method" as one of the three methods compared here seems kind of out of place in this analysis since it is not a method for predicting future PFT distributions as with the DGVM and DM methods. What is the reasoning / purpose behind including RS in this comparison? Or could / should it be used in this study more as a 'reference' data set, like the AR data?

*We understand the concern of Referee #2 on this point. We also agree that RS is often being used as a verification/reference dataset in land surface modelling. However, the emphasis of this work is on improving the DGVM for the 'present-day', based on the premise that the better DGVM are able to predict the present-day distribution of vegetation (based on the processes/parameters driving the DGVM), the more reliable predictions for the future will the model be able to produce. Moreover, RS is also of interest from the perspective that products derived from RS data may also be burdened with uncertainties, needing evaluation - just as DM and DGVM - against a ground-truth/reference data set, which in this case is AR (see also our response to RC-3.5). We will make this clearer in the revised version of the manuscript.*

**Specific Comments**

RC-2.3 - 25-26. please consider this statement carefully; numerous authors could claim that this is untrue

*Thank you for pointing this out. This comment accords with a comment of Referee #3 (RC-3.6) and we will modify this statement in the abstract of the revised manuscript as well as the introduction (lines 83-84) where the amended sentence will be supported by references (e.g., Druel et al 2017)*

RC-2.4 - 34. can these three thresholds be named here, or at least hint at what they are (e.g. ". . . based on . . .)?

*Yes, we agree that the thresholds should be mentioned here. Also, in line with your other comment (RC-2.15), we will adjust the text to clarify that only precipitation seasonality (bioclim_15) is influential.*

RC-2.5 - 115-116. this is not quite clear and perhaps needs to be specified or qualified; i.e. don't many "countries" have national-scale inventory programs?

*This will be re-worded. What is meant here is that wall-to-wall vegetation surveys on national scale are rarely made. AR (the reference dataset) is an example of an area-representative survey.*

RC-2.6 - 126-131. Selecting only 20 plots seems limited, even if deemed acceptable for bioclimatic variation. There needs to be better explanation / justification for this choice, how "acceptable" was determined, and whether a kriging of temperature and precipitation really captures "bioclimatic" variation across the country.

*We agree that a set of 20 plots is a rather limited number. Referee #1 raises the same issue (RC-1.3), and our response (and justification for the choice) is given in comments to Referee #1. We will amend the text to explain our choice better.*

*The representativeness was tested for and explained in supplements S3 and S4 (see also Fig.S3 and Table S4). By acceptable representativeness we mean that the selection of 20 plots does capture the variation across the whole range of temperature and precipitation (in the revised version we will add also "precipitation seasonality" - Fig.S3 – following comment RC-1.3) compared to the full set of 1081 AR plots. The representativeness of the 20 plots was also tested against the full dataset of 1081 AR plots with regard to PFT coverage, where a Chi-square test showed that the two datasets are much more similar than expected by chance.*

*We agree that the sentence on line 131 is not clearly formulated. Also, in line with the comment RC-2.1, we will make sure that it is clear what was done in this study vs. previous studies. Kriging was used in a previous study to interpolate the original SeNorge2 dataset from 1km down to 100m for the purpose of distribution modelling (a procedure which was done and described in Horvath et al. 2019). We agree that this information is not relevant for the representativeness comparison, and it is more important to include a specific description of how the representativeness test was done in this study (in addition to the existing description in supplement S3). We will reformulate this paragraph in the revised manuscript accordingly.*

RC-2.7 - 150. curious decision to give a new acronym to CLM. why not just refer to it as "CLM"? and actually, you do, somewhat, as it seems to switch back-and-forth between "DGVM" and "CLM4.5" for the rest of the manuscript. I see the idea to associate the results from CLM as representative of the "DGVM" approach, but when describing or referring to the specifics of CLM then just call it "CLM" (or "CLM4.5")

*We understand the confusion here. The terms will be further explained. CLM has an option to run will full vegetation dynamics (CLM4.5BGCDV), this option is further referred to as DGVM. The abbreviation of DGVM is used throughout the manuscript to refer to this particular setup of CLM. This will be clarified in the revision, and consistency in the use of terms will be carefully checked.*

RC-2.8 - 154. it may be useful here to point out what these simple assumptions are, and how different (or not) they are from those for which the DM method is based on.

*We will add more details of the assumptions used in DGVM in describing establishment, survival, mortality and light competitions. Compared to DM which uses statistical relationships (line 180) to predict the probability of VTs/PFTs from environmental variables, DGVM assume a simple environmental threshold for establishment, survival and mortality of a PFT to occur (see supplement S6) This will be motivated for and explained made clearly in the revised version of the manuscript.*

RC-2.9 - 171. was soil C initialized somehow, or was it a separate (longer) spin-up? are these mostly undisturbed sites or was that taken into consideration for the vegetation spin-up at each site? was the CORDEX climate used for the spin-up? average or de-trended?

*Thanks for pointing this out. In our experiments, soil C and N were firstly initialized using the restart file from an existing global present-day spin-up simulation with prescribed vegetation. Then, they are spun-up together with vegetation for 400 years.  All the selected sites are mostly undisturbed. The 30-year CORDEX data were cycled during the spin-up. A 30-year period is consistent with WMO climatological normals based on the rational that 30 year is short enough to avoid large long-term trends while long enough to include the range of variability. Thus, the data is not de-trended or averaged. We noticed that vegetation distribution is insensitive to interannual variation or decadal variation of the climate forcing when it reaches equilibrium state in most of our study sites (see supplement S10).  This will be now specified in more detail in this paragraph of the manuscript.*

RC-2.10 - 174. what year / era does this RS map represent? Table 2. I don't think all of this detail is necessary in the main text.

*A very good point, which should be clarified indeed. The RS product used in this study is created from satellite images covering the period of 1999-2006 (Johansen, 2009). We will make this clear in the manuscript.*

*We agree that Table 2 might be too detailed for the main text. We will move Table 2 into the supplement.*

RC-2.11 - 278, 279 & 305 are confusing uses of sub-headings

*We agree that further splitting the chapter 4 (Sensitivity experiments and model improvement) into methods and results might seem untraditional. We suppose that it has not been made clear that the paper falls into two parts: an analysis of data, and a sensitivity analysis which is based upon the results of the analysis. We will add a motivation sentence at the end of the introduction, clearly telling that the sensitivity experiments are a separate chapter, which builds upon the results of the analyses. In that case chapter 4 would remain, but the sub-headings would be removed and instead split into separate paragraphs. (see also reply to RC-3.23)*

RC-2.12 - 287. swe_10 and tmin_5 make sense as described but can "precipitation seasonality" be explained? "bioclim_15" is not as obvious as the other two parameters

*A very good point. We will include a description and a reference to how "precipitation seasonality" is calculated (O'Donnell & Ignizio, 2012). "Precipitation seasonality" is defined as the ratio of the standard deviation of the monthly total precipitation to the mean monthly total precipitation (also known as the coefficient of variation) and is expressed as a percentage.*

RC-2.13 - 293-299. there just seems like so much of the justification and explanation of decisions and approaches for the sensitivity test are glossed over here. For example, why are these particular

parameters chosen, how was bioclim calculated, is the stepwise order important, what does it mean "three PFTs at the same time", how were the thresholds determined, etc etc. Perhaps a little more explanation than just "see Horvath et al 2019" (line 286) would be helpful.

*We agree with the Referee #2. Since a lot of the sensitivity experiments is based on the results from the previous study by Horvath et al. 2019, referring to this article is necessary. However, we agree that explicitly describing the sensitivity experiments is important. We will add more detailed explanation on the reasoning behind the set-up of the sensitivity experiments, including the specific topics that Referee #2 is pointing to in this comment.*

RC-2.14 - 414-415. this seems like a bit of a leap without a more direct connection to the results of this study.

*We agree that the arguments in this paragraph are not supported by the results of this study. In line with the comments from Referee #3 (RC-3.26) and request from Referee #1 (RC-1.7) we decided to remove this argumentation from the revised version of the manuscript.*

RC-2.15 - 468. but in line 312 it was stated that two of those three "had little effect"

*Yes, this must be a remnant of a previous formulation. We will remove the two parameters that did not improve the DGVM performance from this sentence. We will also amend the abstract with regard to this (see also reply to a comment for RC-2.4 and RC-2.17)*

RC-2.16 - 498-499. when are high-quality RS products ever not available anymore in this day-and-age?

*We agree that this needs to be reformulated to explain the challenges clearly. It is not the "high-quality" of RS products in terms of resolution or coverage that we are concerned about, but rather in terms of being able to supply proxies of other properties (such as deriving parameter improvements, traits or in some cases vegetation distribution in high enough thematic resolution). In particular, at high latitudes low sun-angle results in large shadow effects. Furthermore, our results show that analyses of high spatial resolution RS images have limitations when it comes to thematic precision and resolution. We will reformulate this sentence.*

RC-2.17 - 503. Just to be clear, it seems that these parameters were identified in a previous study, not this one, correct? And actually in this study only one of them (bioclim_15) was found to be useful, no? This same claim is made in the abstract, as well, and should be used with care.

*Yes, we agree, and we will carefully re-formulate the sentences with this regard both in the conclusion and abstract. Please see also related comment RC-2.4 and RC-2.15.*

**Technical Corrections**

RC-2.18 - - please review the grammar, wording and sentence structure throughout

*All the technical and wording amendments suggested below will be implemented in the revised version of the manuscript and the text will be carefully searched for erroneous grammar.*

42. please re-word and fix the grammar of this sentence one way or the other

55. remove "the" before DGVMs

60. latitudes

150. replace "further" with "hereafter"

170. "recalculated"

Table 2. "AR" is missing from the caption

292. change "NEB" to "NET", I think

341. "spectre" should be "spectrum"?

412. "overprediction of Boreal NET"?

**3 Anonymous Referee #3**

The manuscript "Improving the representation of high-latitude vegetation in Dynamic Global Vegetation Models" by Horvath et al analyses the performance of three different vegetation modeling approaches with regard to the spatial distribution and relative abundance of plant functional types (PFT) in Norway. The modeling approaches include a dynamic global vegetation model (DGVM), remote sensing (RM), and a statistical distribution model (DM), which relates occurrences of vegetation types to multiple environmental variables. The authors found that both RM and DM showed a better performance than the DGVM when compared to observational data from a range of field sites. They then tested if it was possible to use the DM to improve the predictions of the DGVM with regard to PFT composition and distribution. It was found that, through inclusion of three further bioclimatic constraints based on the analysis of the DM, the performance of the DGVM could be improved. The authors recommend DM as a complementary tool for the assessment and improvement of DGVMs.

RC-3.1 - The manuscript is well written and easy to understand in general. The research topic (assessing and improving DGVMs at high latitudes) is certainly relevant, and the chosen approach is original and seems useful to me. However, the description of the methods needs to be improved, with regard to the chosen statistical approaches, and also the motivation to carry out certain analyses. It often becomes clear only later in the manuscript why a certain method was applied. I therefore recommend minor revisions before a new version of the manuscript may be submitted.

*We thank Referee#3 for a set of thorough comments. We will improve the sections of the manuscript in line with these comments.*

**Comments:**
RC-3.2 - L 28 While the term 'DGVM' is explained at the beginning of the abstract, the term 'distribution model (DM)' is used in this sentence without previous explanation. Please explain shortly in the abstract what a DM is and how it differs from a DGVM, since some readers may not be familiar with the concept.

*Good point. We will add a sentence about the difference between process based (DGVM) and correlative (DM) models.*

RC-3.3 - L 58 Please define or explain in more detail what you mean by 'thematic resolution'. Furthermore, it should be mentioned that recently, specific high-latitude PFTs, such as mosses, for instance, have been added to a number of DGVMs, e.g. Jules (Chadburn et al, 2015, The Cryosphere), JSBACH (Porada et al 2016, The Cryosphere), or ORCHIDEE (Druel et al 2017, Geoscientific Model Development) and several more.

*The term thematic resolution is meant to refer to number of classes (ex. PFTs) in a model. This will be explained in the revised version of the manuscript. Thank you for pointing to these references, we will consider including them as examples in this paragraph in the revised version of our manuscript.*

RC-3.4 - L 60 Three examples are given for the difficulties of DGVMs to simulate extents of high-latitude PFTs correctly. However, I do not see how the underestimation of forest carbon storage by DGVMs relates to this, since this is rather a consequence, and not a reason for the incorrectly predicted extent. Please explain in more detail.

*Good point. The sentence about carbon storage underestimation will be reformulated so that it will be clear that discrepancies in the DGVM have implications on different systems (e.g. carbon storage),*

RC-3.5 - L 71 Please add a short statement to describe in which regard the RS products are not consistent.

*The study by Myers-Smith et al. (2011) reports a mismatch in the spatial resolution between satellite observations and the spatial heterogeneity of vegetation patches in tundra ecosystems. This will be clarified in the introduction. Also, different satellite products produce varying results with regard to vegetation classification (Majasalmi, T. et al. 2018). We will shortly describe these inconsistencies in the manuscript (please, also see RC-2.2).*

RC-3.6 - L 83 At least one study (Druel et al 2017, Geoscientific Model Development), uses site data to assess the DGVM's performance with regard to plant traits. Please be more specific in this regard, and explain what exactly is new in the validation method.

*Yes, we will reformulate this sentence and make clear that our study focuses on evaluation of vegetation distributions between different models/methods. Also, we will mention the study by Druel et al. (2017) as an example of evaluation with field data.*

RC-3.7 - L 121 I do not understand this sentence: If one plot is 0.9 km2 large, then 1081 plots are around 1000 km2, but 18x18 km are only 324 km2. Also, the plots are distributed throughout Norway, so the 18x18 km area has to mean something else. Is it the distance between the plots on a grid which covers Norway? Please explain.

*Thank you for pointing this out! For us who have been working with these data for so long time, it is easy to forget that it is not obvious how they are structured! There is a regular grid covering the whole land area of Norway on which the plots (in total 1081 plots), each with a size of 0.9km2, is placed every 18 km (in latitude) by 18 km (in longitude). This will now be explained in more details in the revised paper.*

RC-3.8 - L 129 To me it seems that low values of temperature and precipitation are underrepresented in the 20 selected plots compared to the full data set. This should be mentioned here briefly and then considered later in the Discussion section.

*We agree that there is a slight underrepresentation in the frequency of plots with the lower values for temperature and precipitation. However, the most important factor was to include plots covering the range of the temperature and precipitation values experienced, which we have succeeded in (Fig S3). We will add a brief description in section 2.3 "Study plots" and in the Discussion.*

RC-3.9 - L 156ff By using the default surface parameter values for CLM, the DGVM may miss some relevant information to correctly predict PFT distribution, compared to RS and DM. Furthermore, by using climate forcing from 1980-2010 and running the DGVM into a steady state with regard to this period, historical climatic effects, which may influence today's PFT distribution are not considered. These points should be mentioned in the Discussion section of the manuscript.

*We understand the concern of the Referee #3 regarding this aspect. In line with replies to the RC-1.5 we will add more detailed discussion on the issues raised in this comment. As to the concern on the usage of the climate forcing data, we indeed overlooked the historical climate effects on vegetation distribution, which response usually lag several years or decades behind climate changes. However, this is considered to have minor impacts on the large biases observed in DGVM (e.g., too much boreal NET and too few shrubs), as historical climate effect (such as cooler temperature in the past) might actually favor more boreal shrub than boreal NET (please, also see our reasoning to comment RC-2.9). We will clarify this in the Discussion.*

RC-3.10 - L 162 Why was the CORDEX data not also used for the DM method? This should be briefly mentioned here.

*In a previous study (Horvath et al. 2019) the authors have created distribution models for vegetation types with a range of predictors (including SeNorgre2 data), where the statistically important predictors were selected in the forward selection procedure. At that point the SeNorge2 was the most reliable climate dataset available for the whole study area. It will be described further in the section 2.4.3. We will add a comment on the choice of climate data sets, including the choice of the CORDEX, respective the seNorge2 dataset.*

RC-3.11 - L 175 Please explain 'supervised' and 'unsupervised' in more detail.

*While with the supervised classification, training data is based on well labeled data from part of the study area, during the unsupervised classification algorithm is only supplied with the number of output classes. 'Supervised' and 'unsupervised' classification methods will be explained in more detail in the revised version of the manuscript.*

RC-3.12 - L 182 the number of explanatory variables (116) is rather high. It should be shortly explained what these are, and why such a large number is necessary for the regression. Even if this information is provided in Horvath et al (2019), it should be summarized here.

*A short description of the explanatory variables (grouped into categories) will be provided in this section of the revised manuscript. Also, a sentence about forward variable selection procedure will be added, to make clear that only a few of the 116 variables were actually included in each final DM.*

RC-3.13 - L 183 It would be good to add a short summary of the evaluation method for the DM here, so the reader can assess the DM better.

*A short summary of the evaluation procedure will be added. Evaluation of each model was carried out using an independent evaluation data set and by calculating the area under the receiver operator curve (AUC), a threshold-independent measure of model performance commonly used in Distribution modelling. AUC can be interpreted as the probability that the model predicts a higher suitability value for a random presence grid cell than for a random absence grid cell (Fielding & Bell, 1997).*

RC-3.14 - L 186 I wonder if, by discarding all other VT except the most probable one, biases in the distribution of the VTs are introduced. Let us assume the logistic regression predicts a certain VT always with a slightly higher probability than a second one; according to the description, only the first VT would occur in the predicted map at all pixels, and all observations of the second one would be discarded, although this VT occurs quite frequently in reality. Please explain this in more detail.

*This is an interesting and intriguing topic. As the Referee #3 rightfully points out, there is a possibility of slight biases in certain regions, for the reason outlined. However, as far as we are aware, this has not yet been closely investigated. We are preparing a manuscript covering this topic in more detail - The results so far suggest that the approach for compiling the wall-to-wall map from 31 DMs, which we also use here, is performing the best out of the tested approaches (Horvath et al., manuscript in prep.). Additionally, as the probability of presence for each VT is predicted separately for each grid-cell, the probability values for every VT varies independently of the probabilities for the other VTs, throughout the study area. Thus, we regard the chance that one VT consistently outperforms another VT over all the grid cells to be negligible.*

RC-3.15 - L 200 I don't understand why an aggregated PFT profile is needed, I thought that the comparison of the 3 modeling approaches and the AR data is done for each of the 20 plots?

*Indeed, the main comparison is between the 3 modelling approaches and AR on each of the 20 plots (this can be found in figure 2 and 3). But besides, it was also worth investigating the overall performance of the tree methods across the study area. In order to do that, we needed the aggregated PFT profiles!*

RC-3.16 - L 208ff This sounds like one comparison was done with the aggregated profiles (one for each method, aggregated over all 20 plots), using the chi-square test. Then, for each of the 20 plots the profiles were compared regarding their dissimilarity. It is not clear to me, why two different statistical methods were used to compare the models (DM, RS, DGVM) to AR.

*We will clarify this in the revised version of the manuscript. The point here is that we wanted to compare the three models (DM, RS, DGVM) to AR both with respect to the overall pattern (represented by the aggregated profiles) and with respect to their performance on each plot; the latter in order to identify the circumstances under which some of the models deviated strongly from the reference. Accordingly, the chi-square test was used to formally test if the models overall deviated from the reference, while the proportional dissimilarity index (which does not come with a statistical test) was calculated to address the purpose of identifying strongly deviating modelling results at plot scale.*

RC-3.17 - L 222 I thought the dissimilarity index was used to assess the similarity between the 3 modeling approaches and the AR data. Why is it then necessary to do a pairwise Wilcoxon-Mann-Whitney test in addition? Please explain the reasons for the chosen statistical approach in a more detailed way.

*Our statistical analyses serve several purposes of which one is to assess the goodness-of-fit of the modeling results to the reference (I.e., to assess their performance); another (which is addressed by the Wilcoxon-*

*Mann-Whitney tests) is to assess the degree to which the models produce pairwise similar differences. We will explain this in the paragraph.*

RC-3.18 - L 230ff As mentioned above (L200), by aggregating the PFT profiles of the 20 plots, differences in profiles between plots are lost. Hence, it is not possible to evaluate the 3 models with respect to the correct prediction of differences in profiles between individual plots. Also, while the AR data (for each plot) can be interpreted as a random sample, it is not clear to me how the model approaches can be consistently included in this Chi-square test. Moreover, the number of elements (6 PFTs) is actually too small for a Chi-square test. The authors need to justify this better, or change their testing approach.

*The mere purpose of analyzing the aggregated profiles is to assess the models' ability to produce overall predictions of PFTs that accord with the PFTs' overall frequency (as given by the reference). We do not see any reason why the chi-square test should not be useful for a contingency table of 6 classes.*

RC-3.19 - L 249 If I understand Fig. 2 correctly, the lines which connect the dots denote the individual plots, which means that for one method (e.g. DGVM), the dissimilarity can be high (1.0), while for another method (e.g. RS) it can be much lower. The result that the goodness of the fit between a given method and AR data depends on the set of chosen plots may point to some underlying systematic deficiencies of each method and should be discussed later.

*Exactly as you describe, the values of dissimilarity index portrayed as dots connected by lines in Fig.2 represent the similarity of each plot between a particular method and the reference dataset AR for that plot. While the individual dissimilarities may be high, we have good reasons to believe that the selection of 20 plots is sufficiently representative for the study area that the major patterns emerging from the analyses reflect real major patterns. Furthermore, you are right that systematic deficiencies in some of the methods are reflected in the single-plot patterns shown in Fig. 2. Some of these were discussed in the previous version of our manuscript and we will carefully search for more when we prepare our revision. These will then be taken into account in the discussion.*

RC-3.20 - L 252 The statement in this sentence is not evident to me in Fig. 3, because this figure simply shows the profiles for each plot (which is a good way of illustrating the results, in my opinion). Wrong reference?

*Absolutely. This typo will be corrected to Fig.2*

RC-3.21 - L 254 Please see also my comment to L 222; I assume that the authors use the Wilcoxon test to assess if the median values of the dissimilarity indices for the 3 models are significantly different from each other. However, I think it is more relevant how the models differ to each other with regard to the AR data. This information is contained in the values of the dissimilarity index, and it should be reported more clearly here. The pairwise comparison of the 3 models seems to me of secondary importance to assess the goodness of the fit to AR data.

*This is correct. The core result we report in this paragraph is the dissimilarity between the methods and the reference dataset. This is reported on lines 249-250 "While RS had the lowest median proportional dissimilarity with the AR reference (0.19, compared to 0.26 for DM and 0.41 for DGVM), …".*

*The pairwise comparison results of the Wilcoxon rank-sum tests are mentioned only after the core findings to support the similarity between RS and DM at most plots. We will ensure that this is clear in the revised paper.*

RC-3.22 - L 262ff The visual comparison of the 3 models in Fig.3 and the associated description is more helpful to assess the modeling approaches than the statistical methods described before.

*In the revised version of the manuscript, we will give more emphasis to the discussion of Fig. 3 in terms of model performance (do a carry out a joint assessment of the figure and the results of the statistical methods).*

RC-3.23 - L 279ff This belongs into the Methods section. Explaining the sensitivity analysis earlier also makes it much easier to understand the goal of the overall approach.

*Please see also our reply to RC-2.11. We will make clear in the introduction, that the sensitivity experiments are a separate chapter, which builds upon the results of the analyses. However, we will delete the subheadings 4.1 Methods and 4.2. Results to avoid confusion.*

RC-3.24 - L 287 The term 'precipitation seasonality' should be better described, in particular since it is found later that it is important to improve DGVM parameterization.

*Please see also our reply to RC-2.12. "Precipitation seasonality" is defined as the ratio of the standard deviation of the monthly total precipitation to the mean monthly total precipitation (also known as the coefficient of variation) and is expressed as a percentage. This will be added to the revised manuscript.*

RC-3.25 - L 379ff The point about 'good' and 'poor' DMs is not clear to me. Why should poor DMs be used at all? Please explain, and also consider my comment above (L 186).

*The terms 'good' and 'poor' refer to the predictive performance of the individual DMs (i.e. AUC - see also reply to comment RC-3.13). The study by Horvath et al. (2019) provides predictions of the distribution of a total of 31 vegetation types across the study area of Norway (with AUC values ranging from 0.671 to 0.989). Reasons for the low predictive performance of some DM may vary, but in this case is most likely caused by missing important predictors. The set of predictor variables used in the study (n=116) might seem excessive, but nevertheless the authors conclude that several important factors are not represented among these 116 (soil nutrients, NDVI, LiDAR etc.). The reason for this is that variables representing these factors were not available in the required formats/resolution/coverage at the time-point the study was carried out; a general problem in distribution modelling. By using the chosen set of predictor variables, statistical approach and settings, the authors obtained the best possible distribution models, even though with regard to the AUC values, some might be considered weak/poor. The direct answer to the comment is that the DM method requires estimates for the probabilities of occurrence for (almost) all vegetation types to create a seamless vegetation map, which in turn is required for making estimates for the PFT profiles as robust as possible. Thus, in this context, 'poor' models are better than no model. We will make this (important) point more clear in the revised version of the manuscript.*

RC-3.26 - L 411 It may not be clear to readers why the lack of a shade-intolerant birch-PFT in the DGVM leads to the over-representation of NET in plots 17 and 18. The birch-PFT should rather have an advantage in mountainous regions compared to NET, which is currently lacking in DGVMs. Please clarify.

*Please see also our reply to RC-1.7 and RC-2.14. We agree with the Referee #3 that this argument is not clear and without a clear support from our results. We will remove the argument from the revised manuscript.*

RC-3.27 - L 450 Please check the literature for the recent progress in including high-latitude vegetation types into the PFT scheme of DGMVs, and add this to the discussion.

*We will study the recent literature on this topic and add more recent references in the discussion. See also our reply to RC-1.8.*

RC-3.28 - L 467 This sentence is hard to understand, please reformulate.

*Yes, this will be reformulated.*

RC-3.29 - L 475 It should be mentioned if increased seasonality promotes or impedes growth of NET.

*Thanks for pointing this out. By applying the new threshold, the growth of NET is impeded if the value for precipitation seasonality is larger than 50 (Table 4, Supplement S6 and S11). This will be mentioned in the revised manuscript.*

**Supplement:**
RC-3.30 - L 40 missing reference L 51 missing reference L 52 missing reference

*Thanks for pointing this out. This is a remnant of splitting the document into manuscript and supplement. All the references are now fixed.*

RC-3.31 - L 55 The PFTs for this study are not in bold font, but shaded grey, please make this consistent.

*This will be fixed*

RC-3.32 - L 56 The caption of Tab. S6 should be a bit more detailed: Is zbot the bottom height of the canopy (11.5 m above ground)? How is the coefficient of variation in precipitation seasonality computed?

*We will adjust the caption to clarify all the mentioned abbreviations*

RC-3.33 - L 90 The cover fractions in plots 801,2108,4268 are clearly not in a steady state. Please check if this significantly affects the results (e.g. by extrapolating the trends in cover), and repeat the DGVM runs, if necessary.

*Thanks for pointing this out. We will extend the running time of our simulations for these sites to check when the vegetation distribution reaches a steady state, and we will investigate whether has an impact on our results.*

RC-3.34 - L 122 missing reference

**Comments on style:**
*All the following comments on style will be implemented in the revised version of the manuscript.*

L 42 I think 'an' is not needed here.

L 55 'DGVMs' instead of 'the DGVMs'

L 60 'at high latitudes' instead of 'in the high latitude'

L 66 'in' not necessary

L 138 the second "of the" is not necessary

L 373 add 'the' before 'reason'

L 401 'differ' instead of 'differs'

**4   REFERENCES:**

Ahti, T., Hämet-Ahti, L. & Jalas, J. 1968. Vegetation zones and their sections in northwestern Europe. – Annls bot. fenn. 5: 169-211.

Bakkestuen, V., Erikstad, L. & Halvorsen, R. 2008. Step-less models for regional environmental variation in Norway. – J. Biogeogr. 35: 1906-1922.

Fielding, A. H., & Bell, J. F., 1997. A review of methods for the assessment of prediction errors in conservation presence/absence models. Environmental Conservation, 24(1), 38–49.

Horvath, P., Halvorsen, R., Stordal, F., Tallaksen, L. M., Tang, H., and Bryn, A.: Distribution modelling of vegetation types based on area frame survey data, Applied Vegetation Science, 22, 547-560,

Johansen, B. E.: Satellittbasert vegetasjonskartlegging for Norge, Direktoratet for Naturforvaltning, Norsk Romsenter, 2009.

Majasalmi, T., Eisner, S., Astrup, R., Fridman, J., and Bright, R. M.: An enhanced forest classification scheme for modeling vegetation–climate interactions based on national forest inventory data, Biogeosciences, 15, 399–412,

O'Donnell, M.S., and Ignizio, D.A., 2012. Bioclimatic predictors for supporting ecological applications in the conterminous United States: U.S. Geological Survey Data Series 691, 10 p

---

## Author Response (AR1)

*Dear Referees,*

*On behalf of all the co-authors I thank you for the insightful and constructive comments directed to the manuscript "Improving the representation of high-latitude vegetation in Dynamic Global Vegetation Models". We have prepared point-by-point responses to each of the comments and amended the manuscript in line with these comments. For convenience and reference, we have numbered the Referee comments with "RC-x.x", where the first "x" corresponds to the referee number and the second "x" to the respective comment. Each of our responses is offered below the respective comment emphasized in blue italics. Please note that the line numbers point to the "marked-up" version of the manuscript attached below the responses.*

*Kind Regards,*

*Peter Horvath*

**Contents**

**1   Anonymous Referee #1**

**General comments**

The overall objective of this paper was to identify biases in a dynamic global vegetation model (DGVM) and, if possible, to find ways of reducing the biases. The analysis focused primarily on relatively undisturbed landscapes in Norway. The target model output was the within-gridcell plant functional type (PFT) distribution. One unique and valuable aspect of the manuscript was that the PFT distributions predicted by the DGVM were compared to multiple products, including field surveys, satellite products, and the output of species distribution models. Field surveys were much more similar to the satellite products and distribution models than to the DGVM. Improvement to the DGVM was realized by incorporation of a precipitation seasonality index, although it was clear that this improvement would not be the end of the story.

Given that PFT distribution is an important quantity that is still challenging for DGVMs to predict, I think that the manuscript covers a topic that will be interesting and useful to readers of Biogeosciences. I also appreciated how the DGVM was compared to multiple products and how the distribution model was leveraged. However, I think that the value of the manuscript could be increased by being more thorough with the methods (see below). Also, I think that more could be done to make the manuscript interesting to readers who use models other than CLM.

*We are thankful to Referee #1 for his/her positive response and constructive comments.*

**Specific comments**

RC-1.1 - The title should be modified. It mentions "Dynamic Global Vegetation Models" in the plural, but only one model is discussed. I also think the title is too general. I would suggestion "high-latitude vegetation distributions" rather than simply "high-latitude vegetation".

*This is a good suggestion. We have adjusted the title to specify that high-latitude vegetation distributions are considered. With regard to the plural mention of DGVMs, we believe that even though we tested this particular exercise only on one DGVM (namely CLM4.5BGCDV), the procedures/methods of implementing variables from DM as new parameters in DGVM can be used in several DGVMs not just the tested one (thus the plural form).*

RC-1.2 - Lines 83-84: This point is overstated. There are publications that have evaluated PFT distributions from dynamic vegetation models against field-based datasets, at least on regional and national scales.

*In line with response to Referee #3 on this same point (see also comment RC-3.6), we have adjusted the formulation of the sentence and added a reference (line 95).*

RC-1.3 - Methods: I am puzzled by the limitation of the study to only 20 plots. Certainly these 20 plots span the range of mean annual temperature and precipitation, but other factors are also commonly perceived to be important. Indeed, the distribution model seems to take 100+ inputs. Some questions that come to mind is whether the plots span the range of observed precipitation seasonality (identified by this study as an important factor!), soil texture, and soil nutrients.

*We agree that a higher number of plots would have been beneficial. Ideally, we would want 1000+ plots or perhaps a regional/global simulation. However, labor-demanding preparation of all data layers for each*

*plot was one of the critical factors for this study and we had to find a compromise between what was practically possible and what was considered robust in terms of the aim of the study. From a methodological perspective, our opinion is clearly that a representative sample of 20 plots is sufficient to demonstrate the differences between the three methods of representing the vegetation distribution.*

*The gradients of precipitation and temperature are known to be among the most influential for vegetation distribution (e.g., Ahti et al. 1968; Bakkestuen et al. 2008), thus we have chosen to include these particular two variables when selecting the 20 plots. However, we also agree with the Referee #1 in the argument that the 20 plots' representativity across the range of precipitation seasonality should be tested (since this is identified as an important factor). We have clarified this more thoroughly, included a comparative test and added a third diagram to the Supplementary Figure S2 (lines 145-161 -chapter 2.3 and Fig S2 lines 20-30 in the supplement). Please also see the response RC-2.6 to Referee #2 with a similar request.*

RC-1.4 - Line 157: Why not assign the observed soil texture to the 20 plots?

*The observed data on the 20 plots unfortunately do not include information about soil texture. The plots were mapped using wall-to-wall vegetation mapping, where only data about the type of vegetation cover are available.*

RC-1.5 - Section 2.4.3: I am concerned that the DGVM and the DM uses different driver data to represent the same phenomenon. For example, does one use SeNorge2 and the other reanalysis to represent precipitation? Does one use observed soil texture and the other "default" soil texture? If so, might differences in inputs account for differences in the DGVM and DM predictions?

*Absolutely. Ideally, we would use the same climate input data for both DM and DGVM. However, there are technical obstacles: DM uses multi-year monthly averaged climate data as input, while DGVM requires 3-hourly meteorological data as the input. SeNorge2 dataset, which is used in DM, has only daily data available, therefore can only be used for DM but not for driving DGVM. For DGVM, we had to use available reanalysis or regional climate model data for present day climate (CORDEX data in this manuscript). To compare the differences between the driving data for DGVM and DM, we have listed mean annual temperature and precipitation for both datasets in the Table S1 and Figure S4 of the supplement (lines 5-8 and 50-55). There are indeed some minor differences between the two sets of driving data, however it is beyond this study to quantify the effect of these differences. We have devoted a paragraph to clarify the potential bias this may imply in the discussion (lines 542 - 550).*

*Soil texture does not come in as an explanatory variable in the DM, whereas DGVM is using soil texture as an important parameter affecting various processes in soil, such as soil temperature, moisture and organic matter decomposition. We have added a comment on the differences between the input data in the paper and discuss its potential implications (lines 542 - 550).*

RC-1.6 - Line 183: Was the DM model previously tuned to these 20 plots? To Norway?

*The DM was not tuned specifically to these 20 plots. The training data for DM included the whole set of 1081 plots (across Norway) at a different thematic resolution (detailed vegetation types instead of PFTs) and at a scale of one point per polygon. Although the 20 plots were included as a subset of the total 1081 plots, we believe the influence is minimal, since they have gone through a spatial and thematic conversion. Moreover, the DM was evaluated with a completely independent dataset.*

RC-1.7 - Line 414: Might phenology also be an issue? Further, what is the light compensation point of the PFTs? Perhaps the authors can use the light compensation point to directly evaluate the relative shade tolerance of the different PFTs.

*Please also see comments to Referee #2 (RC-2.14) and Referee #3 (RC-3.26) regarding this paragraph in the discussion. Phenology is likely to be an issue, as evergreen plants seem to have advantage in competing with deciduous plants in general in the high-latitude region in the model. It is therefore suggested that stress for evergreen plants in winter and spring may not be well represented in the model to limit the growth of boreal NET in some regions. However, we admit that this issue is not well documented through our results and therefore have decided to remove this paragraph from the discussion.*

RC-1.8 - Discussion: Are there lessons for people who use other models? The more the authors can draw out such lessons, the broader the audience this paper would appeal to. The TEM model, which has a more detailed representation of boreal PFT diversity than CLM, immediately comes to mind as one example.

*Thanks for the suggestions. The present-day vegetation distribution outputs from dynamical vegetation models could more often be evaluated by use of multiple products complementing the RS, such as by including DM and AR as presented in this study. We also believe that the procedure of identifying new parameter values from DM, running a set of sensitivity tests and implementing the sensible new parameters into a DGVM is not limited to CLM4.5BGCDV (the DGVM tested here) but transferrable also to other DGVMs, such as the TEM model. We have clarified this and included more thorough discussion with regard to applicability to other models in the revised manuscript (lines 575-579, 614-619).*

**Technical corrections**

RC-1.9 - The manuscript is very readable, but it should still be reviewed for grammar.

*We have carefully searched the manuscript for grammatical errors and corrected where applicable.*

RC-1.10 - Page 3, Lines 43-45: There is a problem with word choice in this sentence. Vegetation distributions are not implemented in ESMs, but rather are predicted by ESMs. The ESM predictions can then be evaluated with satellite products (as done in the present analysis).

*We have rewritten the sentence according to the referee's comment (lines 50-51).*

RC-1.11 - Section 2.4.1: It would be useful for the authors to briefly describe how the DGVM determines the amount of area to each PFT.

*We have added a brief description on how the area of each PFT (i.e. percentage cover fraction %) is determined by DGVM in the revised manuscript (lines 212-217). The percentage cover fraction of each PFT is equal to the average individual's fraction projective cover ($FPC_{ind}$) multiplied by the number of individuals ($N_{ind}$) and average individual's crown area ($CROWN_{ind}$). $FPC_{ind}$ is a function of the maximum leaf carbon achieved in a year, while $CROWN_{ind}$ is related to dead stem carbon simulated by the model. $N_{ind}$ is mainly determined by establishment and survival rate controlled by establishment and survival threshold conditions.*

RC-1.12 - Data availability: Note that the GitHub link not up yet. I understand if the authors do not want to release the link prior to manuscript acceptance, but it is still important not to forget to release the link.

*This is an important point. We have made all the available data accessible on the following repositories (link to DGVM scripts:* https://github.com/huitang-earth/Horvath_etal_BG2020; *link to script for analysis:* https://github.com/geco-nhm/DGVM_RS_DM_Norway; *and link to larger spatial data outputs from RS and DM on DRYAD:* https://doi.org/10.5061/dryad.dfn2z34xn).

**2 Anonymous Referee #2**

**General Comments**

This study evaluates estimates of PFT distributions from a DGVM in comparison to those of remote sensing and empirical models, and against a field-based dataset, for 20 plots of high-latitude vegetation types across Norway. The topic investigated, approach taken, and results reported will be of interest to the modeling community. The paper could benefit from more or better explanation of the methods, especially the CLM simulations. For example, it is unclear whether or not this is intended to be any kind of 'temporally-explicit' analysis; this seems a sort of model estimation of some 'average' PFT distribution from the spin-up results that was compared to field plots and remote sensing data, both of which presumably represent a specific point in time (that is not specified in either case in the methods here).

*Thank you for this to-the-point comment. We agree that more careful explanation of some aspects of the methods is necessary. We have adjusted the manuscript with respect to the specific comments you provided here.*

*This study represents a temporally explicit analysis of the 'present-day' vegetation distribution. We agree and have emphasized this more clearly. In line with further replies to RC-2.10, the temporal context has been specified for each of the three modelling methods as well as for the AR in the respective sub-chapter 2.4 (lines 102, 170, 206-207 and 226).*

RC-2.1 - To properly interpret the results, the sensitivity tests need more explanation and clarification to justify and understand what was done here in this study (vs. previous work).

*We have added a much more detailed explanation of the sensitivity tests in the revised manuscript (chapter 4 - lines 348-397). Also, we shall review the formulations of what was done in this study vs previous work.*

RC-2.2 - The "RS method" as one of the three methods compared here seems kind of out of place in this analysis since it is not a method for predicting future PFT distributions as with the DGVM and DM methods. What is the reasoning / purpose behind including RS in this comparison? Or could / should it be used in this study more as a 'reference' data set, like the AR data?

*We understand the concern of Referee #2 on this point. We also agree that RS is often being used as a verification/reference dataset in land surface modelling. However, the emphasis of this work is on improving the DGVM for the 'present-day', based on the premise that the better DGVM are able to predict the present-day distribution of vegetation (based on the processes/parameters driving the DGVM), the more reliable vegetation predictions for the future will the model be able to produce. Moreover, RS is also of interest from the perspective that products derived from RS data may also be burdened with uncertainties, needing evaluation - just as DM and DGVM - against a ground-truth/reference data set, which in this case is AR (see also our response to RC-3.5). We have devoted lines 79-84 to making this clearer in the revised version of the manuscript.*

**Specific Comments**

RC-2.3 - 25-26. please consider this statement carefully; numerous authors could claim that this is untrue

*Thank you for pointing this out. This comment accords with a comment of Referee #3 (RC-3.6) and we have modified this statement in the abstract of the revised manuscript (lines 25) as well as the introduction (lines 95-96) where the amended sentence is now supported by references (e.g., Druel et al 2017).*

RC-2.4 - 34. can these three thresholds be named here, or at least hint at what they are (e.g. ". . . based on . . .)?*

*Yes, we agree that the thresholds should be mentioned here. Also, in line with another of your comments (RC-2.15), we have adjusted the text to clarify that only precipitation seasonality (bioclim_15) is influential (lines 36-41).*

RC-2.5 - 115-116. this is not quite clear and perhaps needs to be specified or qualified; i.e. don't many "countries" have national-scale inventory programs?

*This has been re-worded (lines 130-131). What is meant here is that wall-to-wall vegetation surveys on national scale are rarely made. AR (the reference dataset) is an example of an area-representative survey.*

RC-2.6 - 126-131. Selecting only 20 plots seems limited, even if deemed acceptable for bioclimatic variation. There needs to be better explanation / justification for this choice, how "acceptable" was determined, and whether a kriging of temperature and precipitation really captures "bioclimatic" variation across the country.

*We agree that a set of 20 plots is a rather limited number. Referee #1 raises the same issue (RC-1.3), and our response (and justification for the choice) is given in comments to Referee #1. We have amended the text to explain our choice better (section 2.3 - lines 145-161).*

*The representativeness was tested for and explained in supplements S2 and S3 (see also Fig.S2 and Table S3 – lines 17-49). By acceptable representativeness we mean that the selection of 20 plots does capture the variation across the whole range of temperature and precipitation (in the revised version we have also added "precipitation seasonality" - Fig.S2 – following comment RC-1.3) compared to the full set of 1081 AR plots. The representativeness of the 20 plots was also tested against the full dataset of 1081 AR plots with regard to PFT coverage, where a Chi-square test showed that the two datasets are much more similar than expected by chance (Supplement S3 – lines 35-49).*

*We have reformulated the sentence on line 131. Also, in line with the comment RC-2.1, we have clarified what was done in this study vs. previous studies. Kriging was used in a previous study to interpolate the original SeNorge2 dataset from 1km down to 100m for the purpose of distribution modelling (a procedure which was done and described in Horvath et al. 2019). We agree that this information is not relevant for the representativeness comparison, and it is more important to include a specific description of how the representativeness test was done in this study (in addition to the existing description in supplement S3). We have reformulated this paragraph and revised manuscript accordingly (section 2.3 - lines 143-161).*

RC-2.7 - 150. curious decision to give a new acronym to CLM. why not just refer to it as "CLM"? and actually, you do, somewhat, as it seems to switch back-and-forth between "DGVM" and "CLM4.5" for the rest of the manuscript. I see the idea to associate the results from CLM as representative of the "DGVM" approach, but when describing or referring to the specifics of CLM then just call it "CLM" (or "CLM4.5")

*We understand the confusion here. This has been clarified and we have explained the terms further in lines 180-182. CLM has an option to run will full vegetation dynamics (CLM4.5BGCDV), this option is further referred to as DGVM. The abbreviation of DGVM is used throughout the manuscript to refer to this particular setup of CLM. Consistency in the use of terms have be carefully checked.*

RC-2.8 - 154. it may be useful here to point out what these simple assumptions are, and how different (or not) they are from those for which the DM method is based on.

*We have added more details about the assumptions used in DGVM in describing establishment, survival, mortality and light competitions (lines 185-186). Compared to DM which uses statistical relationships (line 231-232) to predict the probability of VTs/PFTs from environmental variables, DGVM assume a simple environmental threshold for establishment, survival and mortality of a PFT to occur (see supplement S7).*

RC-2.9 - 171. was soil C initialized somehow, or was it a separate (longer) spin-up? are these mostly undisturbed sites or was that taken into consideration for the vegetation spin-up at each site? was the CORDEX climate used for the spin-up? average or de-trended?

*Thanks for pointing this out. In our experiments, soil C and N were firstly initialized using the restart file from an existing global present-day spin-up simulation with prescribed vegetation. Then, they were spun-up together with vegetation for 400 years. All the selected sites are mostly undisturbed. The 30-year CORDEX data were cycled during the spin-up. A 30-year period is consistent with WMO climatological normals based on the rational that 30 year is short enough to avoid large long-term trends while long enough to include the range of variability. Thus, the data were not de-trended or averaged. We noticed that vegetation distribution was insensitive to interannual variation or decadal variation of the climate forcing when it reached equilibrium state in most of our study sites (see supplement S10). This has now been specified in more detail in the manuscript (section 2.4.1 - lines 195-207).*

RC-2.10 - 174. what year / era does this RS map represent? Table 2. I don't think all of this detail is necessary in the main text.

*A very good point, which should be clarified indeed. The RS product used in this study is created from satellite images covering the period of 1999-2006 (Johansen, 2009). This has been clarified in the manuscript (line 226).*

*We agree that Table 2 might be too detailed for the main text. We have moved Table 2 into the supplement S5 (lines 60-64 in the supplement).*

RC-2.11 - 278, 279 & 305 are confusing uses of sub-headings

*We agree that further splitting the chapter 4 (Sensitivity experiments and model improvement) into methods and results might seem untraditional. We suppose that it has not been made clear that the paper falls into two parts: an analysis of data, and a sensitivity analysis which is based upon the results of the analysis. We have added a motivation sentence at the end of the introduction (line 106), clarifying that the sensitivity experiments are a separate chapter, which builds upon the results of the analyses. Chapter 4 describing the sensitivity experiments has remained, but the sub-headings have been removed and the text into separate paragraphs (lines 342-402) (see also reply to RC-3.23).*

RC-2.12 - 287. swe_10 and tmin_5 make sense as described but can "precipitation seasonality" be explained? "bioclim_15" is not as obvious as the other two parameters

*A very good point. We have now included a description and a reference to how "precipitation seasonality" is calculated (O'Donnell & Ignizio, 2012) on lines 357-359. "Precipitation seasonality" is defined as the ratio of the standard deviation of the monthly total precipitation to the mean monthly total precipitation (also known as the coefficient of variation) and is expressed as a percentage.*

RC-2.13 - 293-299. there just seems like so much of the justification and explanation of decisions and approaches for the sensitivity test are glossed over here. For example, why are these particular parameters chosen, how was bioclim calculated, is the stepwise order important, what does it mean "three PFTs at the same time", how were the thresholds determined, etc etc. Perhaps a little more explanation than just "see Horvath et al 2019" (line 286) would be helpful.

*We agree with the Referee #2. Since a lot of the sensitivity experiments are based on the results from the previous study by Horvath et al. 2019, referring to this article is necessary. However, we agree that explicitly describing the sensitivity experiments is important. We have now added more detailed explanation on the reasoning behind the set-up of the sensitivity experiments, including the specific topics that Referee #2 is pointing to in this comment (lines 342-402).*

RC-2.14 - 414-415. this seems like a bit of a leap without a more direct connection to the results of this study.

*We agree that the arguments in this paragraph are not supported by the results of this study. In line with the comments from Referee #3 (RC-3.26) and request from Referee #1 (RC-1.7) we have removed this argument from the revised version of the manuscript (lines 508-513).*

RC-2.15 - 468. but in line 312 it was stated that two of those three "had little effect"

*Yes, this must be a remnant of a previous formulation. We have removed the two parameters that did not improve the DGVM performance from this sentence (line 587). We have also amended lines 36-41 the abstract with regard to this (see also reply to a comment for RC-2.4 and RC-2.17).*

RC-2.16 - 498-499. when are high-quality RS products ever not available anymore in this day-and-age?

*We agree that this needs to be reformulated to explain the challenges clearly. It is not the "high-quality" of RS products in terms of resolution or coverage that we are concerned about, but rather in terms of being able to supply proxies of other properties (such as deriving parameter improvements, traits or in some cases vegetation distribution in high enough thematic resolution). In particular, at high latitudes low sun-angle results in large shadow effects. Furthermore, our results show that analyses of high spatial resolution RS images have limitations when it comes to thematic precision and resolution. We have now reformulated this sentence (lines 628-629).*

RC-2.17 - 503. Just to be clear, it seems that these parameters were identified in a previous study, not this one, correct? And actually in this study only one of them (bioclim_15) was found to be useful, no? This same claim is made in the abstract, as well, and should be used with care.

*Yes, we agree, and we have carefully re-formulated the sentences with this regard both in the conclusion and abstract. Please see also related comment RC-2.4 and RC-2.15.*

**Technical Corrections**
RC-2.18 - - please review the grammar, wording and sentence structure throughout

*All the technical and wording amendments suggested below have been implemented in the revised version of the manuscript. The text has been carefully searched and corrected for erroneous grammar.*

42. please re-word and fix the grammar of this sentence one way or the other

55. remove "the" before DGVMs

60. latitudes

150. replace "further" with "hereafter"

170. "recalculated"

Table 2. "AR" is missing from the caption

292. change "NEB" to "NET", I think

341. "spectre" should be "spectrum"?

412. "overprediction of Boreal NET"?

**3   Anonymous Referee #3**

The manuscript "Improving the representation of high-latitude vegetation in Dynamic Global Vegetation Models" by Horvath et al analyses the performance of three different vegetation modeling approaches with regard to the spatial distribution and relative abundance of plant functional types (PFT) in Norway. The modeling approaches include a dynamic global vegetation model (DGVM), remote sensing (RM), and a statistical distribution model (DM), which relates occurrences of vegetation types to multiple environmental variables. The authors found that both RM and DM showed a better performance than the DGVM when compared to observational data from a range of field sites. They then tested if it was possible to use the DM to improve the predictions of the DGVM with regard to PFT composition and distribution. It was found that, through inclusion of three further bioclimatic constraints based on the analysis of the DM, the performance of the DGVM could be improved. The authors recommend DM as a complementary tool for the assessment and improvement of DGVMs.

RC-3.1 - The manuscript is well written and easy to understand in general. The research topic (assessing and improving DGVMs at high latitudes) is certainly relevant, and the chosen approach is original and seems useful to me. However, the description of the methods needs to be improved, with regard to the chosen statistical approaches, and also the motivation to carry out certain analyses. It often becomes clear only later in the manuscript why a certain method was applied. I therefore recommend minor revisions before a new version of the manuscript may be submitted.

*We thank Referee#3 for a set of thorough comments. We have improved the sections of the manuscript in line with these comments.*

**Comments:**

RC-3.2 - L 28 While the term 'DGVM' is explained at the beginning of the abstract, the term 'distribution model (DM)' is used in this sentence without previous explanation. Please explain shortly in the abstract what a DM is and how it differs from a DGVM, since some readers may not be familiar with the concept.

*Good point. We have added a sentence about the difference between process based (DGVM) and correlative (DM) models (lines 29-31).*

RC-3.3 - L 58 Please define or explain in more detail what you mean by 'thematic resolution'. Furthermore, it should be mentioned that recently, specific high-latitude PFTs, such as mosses, for instance, have been added to a number of DGVMs, e.g. Jules (Chadburn et al, 2015, The Cryosphere), JSBACH (Porada et al 2016, The Cryosphere), or ORCHIDEE (Druel et al 2017, Geoscientific Model Development) and several more.

*The term thematic resolution is meant to refer to number of classes (ex. PFTs) in a model. This has now been explained in line 66. Thank you for pointing to these references, we have included them as examples in this paragraph (line 64).*

RC-3.4 - L 60 Three examples are given for the difficulties of DGVMs to simulate extents of high-latitude PFTs correctly. However, I do not see how the underestimation of forest carbon storage by DGVMs relates to this, since this is rather a consequence, and not a reason for the incorrectly predicted extent. Please explain in more detail.

*Good point. The sentence about carbon storage underestimation has been reformulated (line 68) to clarify that discrepancies in the DGVM have implications on different systems (e.g. carbon storage).*

RC-3.5 - L 71 Please add a short statement to describe in which regard the RS products are not consistent.

*The study by Myers-Smith et al. (2011) reports a mismatch in the spatial resolution between satellite observations and the spatial heterogeneity of vegetation patches in tundra ecosystems. This will be clarified in the introduction. Also, different satellite products produce varying results with regard to vegetation classification (Majasalmi, T. et al. 2018). We have devoted lines 80-81 to describing these inconsistencies in the manuscript (please, also see RC-2.2).*

RC-3.6 - L 83 At least one study (Druel et al 2017, Geoscientific Model Development), uses site data to assess the DGVM's performance with regard to plant traits. Please be more specific in this regard, and explain what exactly is new in the validation method.

*Yes, we have reformulated this sentence to make clear that our study focuses on evaluation of vegetation distributions between different models/methods (lines 94-99). Also, we mention the study by Druel et al. (2017) as an example of evaluation with field data.*

RC-3.7 - L 121 I do not understand this sentence: If one plot is 0.9 km2 large, then 1081 plots are around 1000 km2, but 18x18 km are only 324 km2. Also, the plots are distributed throughout Norway, so the 18x18 km area has to mean something else. Is it the distance between the plots on a grid which covers Norway? Please explain.

*Thank you for pointing this out! For us, having worked with these data for so long time, it is easy to forget that it is not obvious how they are structured! There is a regular grid covering the whole land area of*

*Norway on which the plots (in total 1081 plots), each with a size of 0.9km2, is placed every 18 km (in latitude) by 18 km (in longitude). This has now been explained in more details in lines 136-139.*

RC-3.8 - L 129 To me it seems that low values of temperature and precipitation are underrepresented in the 20 selected plots compared to the full data set. This should be mentioned here briefly and then considered later in the Discussion section.

*We agree that there is a slight underrepresentation in the frequency of plots with the lower values for temperature and precipitation. However, the most important factor was to include plots covering the range of the temperature and precipitation values experienced, which we have succeeded in (Fig S3). We have added a brief description in lines 155-156.*

RC-3.9 - L 156ff By using the default surface parameter values for CLM, the DGVM may miss some relevant information to correctly predict PFT distribution, compared to RS and DM. Furthermore, by using climate forcing from 1980-2010 and running the DGVM into a steady state with regard to this period, historical climatic effects, which may influence today's PFT distribution are not considered. These points should be mentioned in the Discussion section of the manuscript.

*We understand the concern of the Referee #3 regarding this aspect. In line with replies to the RC-1.5 we have added a more detailed discussion on the issues raised in this comment (lines 542-550). As to the concern on the usage of the climate forcing data, we indeed overlooked the historical climate effects on vegetation distribution, which usually lag several years or decades behind climate changes. However, this is considered to have minor impacts on the large biases observed in DGVM (e.g., too much boreal NET and too few shrubs), even though historical climate effects (such as cooler temperature in the past) might favor more boreal shrub than boreal NET (please, also see our reasoning to comment RC-2.9). We have devoted a paragraph to clarify this in the Discussion (lines 536-542).*

RC-3.10 - L 162 Why was the CORDEX data not also used for the DM method? This should be briefly mentioned here.

*In a previous study (Horvath et al. 2019) the authors have created distribution models for vegetation types with a range of predictors (including SeNorgre2 data), where the statistically important predictors were selected in a forward selection procedure. At that point the SeNorge2 was the most reliable climate dataset available for the whole study area. We have now added a comment on the choice of climate data sets in DM in the section 2.4.3 (lines 235-237). Also see the paragraph in discussion on lines 536-642.*

RC-3.11 - L 175 Please explain 'supervised' and 'unsupervised' in more detail.

*While in supervised classification, training data are based on well labeled data from part of the study area, unsupervised classification is only supplied with the number of output classes. 'Supervised' and 'unsupervised' classification methods are now shortly explained on lines 226-228.*

RC-3.12 - L 182 the number of explanatory variables (116) is rather high. It should be shortly explained what these are, and why such a large number is necessary for the regression. Even if this information is provided in Horvath et al (2019), it should be summarized here.

*We have added a short description of the explanatory variables (grouped into categories) on lines 232-233. Also, a sentence about forward variable selection procedure has been added, to make clear that only a few of the 116 variables were actually included in each final DM (lines 233-237).*

RC-3.13 - L 183 It would be good to add a short summary of the evaluation method for the DM here, so the reader can assess the DM better.

*We have now added a short summary of the evaluation procedure on lines 237-242. Evaluation of each model was carried out using an independent evaluation data set and by calculating the area under the receiver operator curve (AUC), a threshold-independent measure of model performance commonly used in Distribution modelling. AUC can be interpreted as the probability that the model predicts a higher suitability value for a random presence grid cell than for a random absence grid cell (Fielding & Bell, 1997).*

RC-3.14 - L 186 I wonder if, by discarding all other VT except the most probable one, biases in the distribution of the VTs are introduced. Let us assume the logistic regression predicts a certain VT always with a slightly higher probability than a second one; according to the description, only the first VT would occur in the predicted map at all pixels, and all observations of the second one would be discarded, although this VT occurs quite frequently in reality. Please explain this in more detail.

*This is an interesting and intriguing topic. As the Referee #3 rightfully points out, there is a possibility of slight biases in certain regions, for the reason outlined. However, as far as we are aware, this has not yet been closely investigated. We are preparing a manuscript covering this topic in more detail - The results so far suggest that the approach for compiling the wall-to-wall map from 31 DMs, which we also use here, is performing the best out of the tested approaches (Horvath et al., manuscript in prep.). Additionally, as the probability of presence for each VT is predicted separately for each grid-cell, the probability values for every VT varies independently of the probabilities for the other VTs, throughout the study area. Thus, we regard the chance that one VT consistently outperforms another VT over all the grid cells to be negligible. We have now explained this more carefully in the discussion (lines 477-482).*

RC-3.15 - L 200 I don't understand why an aggregated PFT profile is needed, I thought that the comparison of the 3 modeling approaches and the AR data is done for each of the 20 plots?

*Indeed, the main comparison is between the 3 modelling approaches and AR on each of the 20 plots (this can be found in figure 2 and 3). But besides, it was also worth investigating the overall performance of the three methods across the study area. In order to do that, we needed the aggregated PFT profiles. We have now clarified this in the sentence (line 259).*

RC-3.16 - L 208ff This sounds like one comparison was done with the aggregated profiles (one for each method, aggregated over all 20 plots), using the chi-square test. Then, for each of the 20 plots the profiles were compared regarding their dissimilarity. It is not clear to me, why two different statistical methods were used to compare the models (DM, RS, DGVM) to AR.

*The point here is that we wanted to compare the three models (DM, RS, DGVM) to AR both with respect to the overall pattern (represented by the aggregated profiles) and with respect to their performance on each plot; the latter in order to identify the circumstances under which some of the models deviated strongly from the reference. Accordingly, the chi-square test was used to formally test if the models overall deviated from the reference, while the proportional dissimilarity index (which does not come with a statistical test) was calculated to address the purpose of identifying strongly deviating modelling results at plot scale. This is now clarified in lines 267-271.*

RC-3.17 - L 222 I thought the dissimilarity index was used to assess the similarity between the 3 modeling approaches and the AR data. Why is it then necessary to do a pairwise Wilcoxon-Mann-Whitney test in addition? Please explain the reasons for the chosen statistical approach in a more detailed way.

*Our statistical analyses serve several purposes of which one is to assess the goodness-of-fit of the modeling results to the reference (I.e., to assess their performance); another (which is addressed by the Wilcoxon-Mann-Whitney tests) is to assess the degree to which the models produce pairwise similar differences. We have added a sentence to explain this in the paragraph (lines 283-284).*

RC-3.18 - L 230ff As mentioned above (L200), by aggregating the PFT profiles of the 20 plots, differences in profiles between plots are lost. Hence, it is not possible to evaluate the 3 models with respect to the correct prediction of differences in profiles between individual plots. Also, while the AR data (for each plot) can be interpreted as a random sample, it is not clear to me how the model approaches can be consistently included in this Chi-square test. Moreover, the number of elements (6 PFTs) is actually too small for a Chi-square test. The authors need to justify this better, or change their testing approach.

*The mere purpose of analyzing the aggregated profiles is to assess the models' ability to produce overall predictions of PFTs that accord with the PFTs' overall frequency (as given by the reference). We do not see any reason why the chi-square test should not be useful for a contingency table of 6 classes.*

RC-3.19 - L 249 If I understand Fig. 2 correctly, the lines which connect the dots denote the individual plots, which means that for one method (e.g. DGVM), the dissimilarity can be high (1.0), while for another method (e.g. RS) it can be much lower. The result that the goodness of the fit between a given method and AR data depends on the set of chosen plots may point to some underlying systematic deficiencies of each method and should be discussed later.

*Exactly as you describe, the values of dissimilarity index portrayed as dots connected by lines in Fig.2 represent the similarity of each plot between a particular method and the reference dataset AR for that plot. While the individual dissimilarities may be high, we have good reasons to believe that the selection of 20 plots is sufficiently representative for the study area that the major patterns emerging from the analyses reflect real major patterns. Furthermore, you are right that systematic deficiencies in some of the methods are reflected in the single-plot patterns shown in Fig. 2. Some of these topics were discussed in the previous version of our manuscript and we have now expanded on some of the aspects (sections 5.1.1., 5.1.2., 5.1.4., and section 5.1.5 of the discussion chapter).*

RC-3.20 - L 252 The statement in this sentence is not evident to me in Fig. 3, because this figure simply shows the profiles for each plot (which is a good way of illustrating the results, in my opinion). Wrong reference?

*Absolutely. This typo has been corrected to Fig.2.*

RC-3.21 - L 254 Please see also my comment to L 222; I assume that the authors use the Wilcoxon test to assess if the median values of the dissimilarity indices for the 3 models are significantly different from each other. However, I think it is more relevant how the models differ to each other with regard to the AR data. This information is contained in the values of the dissimilarity index, and it should be reported more clearly here. The pairwise comparison of the 3 models seems to me of secondary importance to assess the goodness of the fit to AR data.

*This is correct. The core result we report in this paragraph is the dissimilarity between the methods and the reference dataset. This is reported on lines 311-315 "While RS had the lowest median proportional dissimilarity with the AR reference (0.19, compared to 0.26 for DM and 0.41 for DGVM), …".*

*The pairwise comparison results of the Wilcoxon rank-sum tests are mentioned only after the core findings to support the similarity between RS and DM at most plots (lines 315-319).*

RC-3.22 - L 262ff The visual comparison of the 3 models in Fig.3 and the associated description is more helpful to assess the modeling approaches than the statistical methods described before.

*The paragraph on lines 325-336 summarizes the visual inspection of Fig. 3 in terms of performance of the three methods and describes the regional deviations of DGVM and DM from the reference. The issues are further discussed on lines 446-471 and in paragraph 5.1.5.*

RC-3.23 - L 279ff This belongs into the Methods section. Explaining the sensitivity analysis earlier also makes it much easier to understand the goal of the overall approach.

*We agree. Please see also our reply to RC-2.11. We have added a clarification in the introduction (line 106), that the sensitivity experiments are described in a separate chapter, which builds upon the results of the analyses. We have deleted the subheadings 4.1 Methods and 4.2. Results to avoid confusion.*

RC-3.24 - L 287 The term 'precipitation seasonality' should be better described, in particular since it is found later that it is important to improve DGVM parameterization.

*Please see also our reply to RC-2.12. "Precipitation seasonality" is defined as the ratio of the standard deviation of the monthly total precipitation to the mean monthly total precipitation (also known as the coefficient of variation) and is expressed as a percentage. This has now been explained on lines 357 -359.*

RC-3.25 - L 379ff The point about 'good' and 'poor' DMs is not clear to me. Why should poor DMs be used at all? Please explain, and also consider my comment above (L 186).

*The terms 'good' and 'poor' refer to the predictive performance of the individual DMs (i.e. AUC - see also reply to comment RC-3.13). The study by Horvath et al. (2019) provides predictions of the distribution of a total of 31 vegetation types across the study area of Norway (with AUC values ranging from 0.671 to 0.989). Reasons for the low predictive performance of some DM may vary, but in this case is most likely caused by missing important predictors. The set of predictor variables used in the study (n=116) might seem excessive, but nevertheless the authors conclude that several important factors are not represented among these 116 (soil nutrients, NDVI, LiDAR etc.). The reason for this is that variables representing these factors were not available in the required formats/resolution/coverage at the time-point the study was carried out; a general problem in distribution modelling. By using the chosen set of predictor variables, statistical approach and settings, the authors obtained the best possible distribution models, even though with regard to the AUC values, some might be considered weak/poor. The direct answer to the comment is that the DM method requires estimates for the probabilities of occurrence for (almost) all vegetation types to create a seamless vegetation map, which in turn is required for making estimates for the PFT profiles as robust as possible. Thus, in this context, 'poor' models are better than no model. We have devoted the paragraph on (lines 455-465) to making this (important) point more clear.*

RC-3.26 - L 411 It may not be clear to readers why the lack of a shade-intolerant birch-PFT in the DGVM leads to the over-representation of NET in plots 17 and 18. The birch-PFT should rather have an advantage in mountainous regions compared to NET, which is currently lacking in DGVMs. Please clarify.

*Please see also our reply to RC-1.7 and RC-2.14. We agree with the Referee #3 that this argument is not clear and without a clear support from our results. We have decided to remove the argument from the revised manuscript.*

RC-3.27 - L 450 Please check the literature for the recent progress in including high-latitude vegetation types into the PFT scheme of DGMVs, and add this to the discussion.

*We have added recent studies about this topic in the discussion (lines 572-575). See also our reply to RC-1.8.*

RC-3.28 - L 467 This sentence is hard to understand, please reformulate.

*Yes, this has been reformulated.*

RC-3.29 - L 475 It should be mentioned if increased seasonality promotes or impedes growth of NET.

*Thanks for pointing this out. By applying the new threshold, the growth of NET is impeded if the value for precipitation seasonality is larger than 50 (Table 4, Supplement S6 and S11). This is now mentioned in the lines 597-598.*

**Supplement:**

RC-3.30 - L 40 missing reference L 51 missing reference L 52 missing reference

*Thanks for pointing this out. This is a remnant of splitting the document into manuscript and supplement. All the references are now fixed.*

RC-3.31 - L 55 The PFTs for this study are not in bold font, but shaded grey, please make this consistent.

*This has been fixed.*

RC-3.32 - L 56 The caption of Tab. S6 should be a bit more detailed: Is zbot the bottom height of the canopy (11.5 m above ground)? How is the coefficient of variation in precipitation seasonality computed?

*We have adjusted the caption to clarify all the mentioned abbreviations.*

RC-3.33 - L 90 The cover fractions in plots 801,2108,4268 are clearly not in a steady state. Please check if this significantly affects the results (e.g. by extrapolating the trends in cover), and repeat the DGVM runs, if necessary.

*Thanks for pointing this out. We have now extended the running time of our simulations for these three plots by 400, 200 and 200 years respectively to check the vegetation distribution at the equilibrium state. We found that the average of the last 20 years at the end of each simulation does not deviate substantially from our previous results at the end of 400 years (see the new plots added in the supplement S11.2). We have therefore decided to consistently use the original 400 year spin-up data for the analysis for all 20 plots. This is also clarified in lines 204-207).*

RC-3.34 - L 122 missing reference

Comments on style:

*All the following comments on style have been implemented in the revised version of the manuscript.*

L 42 I think 'an' is not needed here.

L 55 'DGVMs' instead of 'the DGVMs'

L 60 'at high latitudes' instead of 'in the high latitude'

L 66 'in' not necessary

L 138 the second "of the" is not necessary

L 373 add 'the' before 'reason'

L 401 'differ' instead of 'differs'

range of variation for a third variable (precipitation seasonality) which was later selected for sensitivity
experiments (see further section 4). While low values of temperature and precipitation are slightly
underrepresented in the 20 plots, the total range of vluesariation was well covered. AllNone of the the tests for
temperature, precipitation and for the additional variable of (precipitation seasonality) do not indicate that

[revised manuscript text omitted]

For each of temperature,  precipitation and precipitation seasonality, we obtained  values for the centrepoint of each AR18×18 plot (cf. Fig. S1) and compared the frequency distributions of the selected plots with those of all plots (Fig. S3). A series of Kolmogorov-Smirnov tests for these three variables (comparison of sample mean and variance) indicate that the subsample does not deviate from the full dataset substantially. The 20 selected plots span elevations from 88 to 1670 m a.s.l., covers an annual temperature range from -4°C to 7.1°C, and an annual precipitation range from 466 to 2661 mm (Fig. S1), which accords well with the variation in the AR18×18

dataset (Fig. S2).

[Figure]

**Figure S2– Frequency distributions of plots in the original AR18×18 dataset (n=1081; in red) and in the set of 20 plots**
**selected for this study (in blue), with respect to annual mean temperature (top left),  annual precipitation (top right)**
**and precipitation seasonality (bottom left). Dashed lines indicate means for the respective datasets.**

Supplement S3     – **Assessment of the representativeness of PFT profiles**

[revised manuscript text omitted]

plot 14

Plant Functional Types
- BG - bare ground
- boreal NET - needleleaf evergreen boreal tree
- temperate BDT - broadleaf deciduous temperate tree
- boreal BDT - broadleaf deciduous boreal tree
- boreal BDS - broadleaf deciduous boreal shrub
- C3 - C3 Grass
- excluded

1000     2000 m

**Figure S6 – Sampling design used by the remote sensing (RS) and distribution modelling (DM) methods and to obtain the AR reference dataset. Like DGVM plots (see Fig. S7), the RS and DM plots are 1×1 km, while the AR plots are 1.5×0.6 km. Plots 7 and plot 14 (AR18x18 plot #1322 and plot #2425) are used as examples.**

Supplement S7 **– DGVM parameters for PFTs (CLM4.5-BGCDV)**

**Table  S7 – Some important PFT parameter settings for DGVM (CLM4.5-DV). PFTs relevant for the study area (Norway) are shaded grey. Prescribed heights for the**
**canopy are indicated by the upper and lower limits in columns "ztop" and "zbot" respectively. Limiting temperatures for survival and establishment are mentioned in columns "Tc,min"**
**and "Tc,max" respectively. Minimum growing degree days for establishment are contained for relevant PFTs in column "GDDmin". The last three columns contain the adjusted**
**parameter thresholds used in the sensitivity experiment. swe_10 – snow water equivalent in October (mm);**
**tmin_5 – minimum temperature in May (°C) bioclim_15 – precipitation seasonality (coefficient of variation);**

[revised manuscript text omitted]

For plots #801, #2108 and #4268, the spin-up was extended by additional 400, 200 and 200 years respectively.

[Figure]

[Figure]

[Figure]

[Figure]

**Figure**  **S11.1 – DGVM spin-up for 400 years and simulation of PFT profiles for each of the 20 plots used in this study. FPCGRID – estimated percentage per PFT per grid cell. Reference number of plots accords with the AR18x18 dataset, and plot numbers can be found in Table S1.**

[Figure]

[Figure]

**Figure S11.2 – Three plots (number 6, 12, 17) where DGVM spin-up was prolonged beyond 400 years and simulation of PFTs was extended by 400, 200 and 200 years respectively in order to check for equilibrium. FPCGRID – estimated percentage per PFT per grid cell. Reference number of plots accords with the AR18x18 dataset, and plot numbers can be found in Table S1**

 Supplement S12 **– Sensitivity experiments: frequency-of-presence (FoP) plots**

Frequency-of-presence (FOP) plots based upon output from distribution models (DM) for the nine combinations of three environmental variables and three vegetation types modelled, used to indicate threshold values that were explored in the sensitivity experiments, are shown in Fig. S11. Thresholds for new variables in DGVM models were chosen based upon visual inspection of the FoP plots. For example, while boreal BDS are abundant below swe_10 value of 380mm, boreal BDT and boreal NET are abundant at values of swe_10 below 180mm and 150mm respectively. Also, while we identified no clear threshold of variable bioclim_15 for boreal BDS and BDT (frequency of presence is never zero along the variable x-axis - lower left and middle panel of Fig S12), threshold for boreal NET was set to 50 (a value above which no presences occur - lower right panel of Fig S12).

[Figure]

**Figure  S12 – Frequency-of-presence plots from the distribution modelling (DM) study by Horvath et al. (2019) for the combinations of environmental predictors and vegetation types (VTs) used in the sensitivity experiments with DGVM. FOP is the frequency of 100×100 m pixels in the AR18×18 dataset in which the VT in question is present, expressed as a fraction of all pixels in that interval along the environmental variable. All environmental variables were a priori divided into 100 intervals with the same number of pixels. The environmental gradients were: swe_10 – snow water equivalent in October (mm); tmin_5 –- minimum temperature in May (°C); bioclim_15 – precipitation seasonality (unitless index). Boreal BDS – boreal broadleaf deciduous shrubs, Boreal BDT - boreal broadleaf deciduous trees, Boreal NET - boreal needleleaf evergreen shrubs.**

Table S12 S13 – PFT profiles for the six out of the 20 plots (plot numbers 1, 2, 5, 15, 17, 18) which were included in the sensitivity experiments, for four 'generations' of DGVM parameter settings and the AR reference dataset. From left to right the column represent: DGVM before adjustment of parameters thresholds; DGVM_adj1 after adjustment first adding parameter threshold of swe_10; DGVM_adj2 after adjustment also adding parameter threshold of tmin_5; DGVM_adj3 after finally adding parameter threshold adjustment of bioclim_15; and the PFT profile of the reference dataset AR. All parameter thresholds were added cumulatively. Full names for the PFTs are given in Table S6 S7 and names of parameters and their values in Table 3.

| | DGVM | DGVM adj1 | DGVM adj2 | DGVM adj3 | AR | DGVM | DGVM adj1 | DGVM adj2 | DGVM adj3 | AR | DGVM | DGVM adj1 | DGVM adj2 | DGVM adj3 | AR | DGVM | DGVM adj1 | DGVM adj2 | DGVM adj3 | AR | DGVM | DGVM adj1 | DGVM adj2 | DGVM adj3 | AR | DGVM | DGVM adj1 | DGVM adj2 | DGVM adj3 | AR |
|---|---|---|---|---|---|---|---|---|---|---|---|---|---|---|---|---|---|---|---|---|---|---|---|---|---|---|---|---|---|---|
| | plot 1 | | | | | plot 2 | | | | | plot 5 | | | | | plot 15 | | | | | plot 17 | | | | | plot 18 | | | | |
| BG | 5 | 5 | 5 | 9 | 0 | 6 | 5 | 5 | 5 | 4 | 6 | 6 | 6 | 7 | 0 | 5 | 5 | 5 | 3 | 13 | 28 | 100 | 100 | 100 | 0 | 5 | 100 | 100 | 100 | 0 |
| boreal NET | 95 | 95 | 95 | 0 | 0 | 58 | 58 | 58 | 0 | 0 | 52 | 52 | 52 | 0 | 0 | 92 | 92 | 92 | 0 | 0 | 72 | 0 | 0 | 0 | 0 | 95 | 0 | 0 | 0 | 0 |
| temp. BDT | 0 | 0 | 0 | 0 | 0 | 2 | 2 | 2 | 33 | 0 | 4 | 4 | 4 | 13 | 0 | 0 | 0 | 0 | 1 | 0 | 0 | 0 | 0 | 0 | 0 | 0 | 0 | 0 | 0 | 0 |
| boreal BDT | 0 | 0 | 0 | 0 | 35 | 2 | 2 | 2 | 31 | 12 | 4 | 4 | 4 | 13 | 0 | 0 | 0 | 0 | 2 | 0 | 0 | 0 | 0 | 0 | 66 | 0 | 0 | 0 | 0 | 70 |
| boreal BDS | 0 | 0 | 0 | 91 | 63 | 32 | 32 | 32 | 31 | 75 | 35 | 35 | 35 | 67 | 99 | 3 | 3 | 3 | 89 | 83 | 0 | 0 | 0 | 0 | 34 | 0 | 0 | 0 | 0 | 30 |
| C3 | 0 | 0 | 0 | 0 | 1 | 0 | 0 | 0 | 0 | 9 | 0 | 0 | 0 | 0 | 1 | 0 | 0 | 0 | 6 | 5 | 0 | 0 | 0 | 0 | 0 | 0 | 0 | 0 | 0 | 0 |

---

## Author Response (AR2)

Peter Horvath
University of Oslo
P.O. Box 1172,
0318 Oslo, Norway
peter.horvath@nhm.uio.no

Associate Editor
*Biogeosciences, Copernicus Publications*

Dear Dr Akihiko Ito,

On behalf of all the co-authors I would like to thank you for evaluating our revised version of the manuscript. Following your suggestions about technical corrections, we have now adjusted the reference style to match the one required by the journal, accessed through:

https://www.biogeosciences.net/Copernicus_Publications.ens

We have also enlarged font size in the requested figures (Fig. 2, S2 and S12) in the manuscript and the supplement.

Kind Regards,

Peter Horvath